# Is modular control related to functional outcomes in individuals with knee osteoarthritis and following total knee arthroplasty?

Rebekah R. Koehn[1], Sarah A. Roelker[2], Xueliang Pan[3], Laura C. Schmitt[4,5,6], Ajit M. W. Chaudhari[1,4,5,6,7,8], Robert A. Siston[1,4,5,7,8]*

1 Department of Mechanical and Aerospace Engineering, The Ohio State University, Columbus, Ohio, United States of America, 2 Department of Kinesiology, University of Massachusetts Amherst, Amherst, Massachusetts, United States of America, 3 Center for Biostatistics and Bioinformatics, The Ohio State University, Columbus, Ohio, United States of America, 4 School of Health and Rehabilitation Sciences, The Ohio State University, Columbus, Ohio, United States of America, 5 Sports Medicine Research Institute, The Ohio State University Wexner Medical Center, Columbus, Ohio, United States of America, 6 Division of Physical Therapy, School of Health and Rehabilitation Sciences, The Ohio State University, Columbus, Ohio, United States of America, 7 Department of Biomedical Engineering, The Ohio State University, Columbus, Ohio, United States of America, 8 Department of Orthopaedics, The Ohio State University, Columbus, Ohio, United States of America

* siston.1@osu.edu

## Abstract

### Background

Individuals who undergo total knee arthroplasty (TKA) for treatment of knee osteoarthritis often experience suboptimal outcomes. Investigation of neuromuscular control strategies in these individuals may reveal factors that contribute to these functional deficits. The purpose of this pilot study was to determine the relationship between patient function and modular control during gait before and after TKA.

### Methods

Electromyography data from 36 participants (38 knees) were collected from 8 lower extremity muscles on the TKA-involved limb during ≥5 over-ground walking trials before (n = 30), 6-months after (n = 26), and 24-months after (n = 13) surgery. Muscle modules were estimated using non-negative matrix factorization. The number of modules was determined from 500 resampled trials.

### Results

A higher number of modules was related to better performance-based and patient-reported function before and 6-months after surgery. Participants with organization similar to healthy, age-matched controls trended toward better function 24-months after surgery, though these results were not statistically significant. We also observed plasticity in the participants' modular control strategies, with 100% of participants who were present before and 24-months

**Data Availability Statement:** Data are available on Figshare. Demographics data for all participants (. csv) and raw EMG, marker, and force plate data (.

mat) for the participants before and after total knee arthroplasty are available at: https://figshare.com/projects/Is_modular_control_related_to_functional_outcomes_in_individuals_with_knee_osteoarthritis_and_following_total_knee_arthroplasty_/132959 Raw EMG, marker, and force plate data (.mat) for the healthy participants are available at: https://figshare.com/projects/Modular_Control_of_Walking_in_Unimpaired_Younger_and_Older_Adults_and_Individuals_with_Knee_Osteoarthritis/128774.

**Funding:** This research was supported by the National Science Foundation Graduate Research Fellowship Program (https://www.nsfgrfp.org/) under Grant Nos. DGE-1343012 (SAR) and DGE-0822215 (EJC), a fellowship from The Ohio State University (RRK; https://gradsch.osu.edu/fellowships), and by Grant Number R01AR056700 from the National Institute of Arthritis and Musculoskeletal and Skin Diseases (https://www-niams-nih-gov.proxy.lib.ohio-state.edu/). The content is solely the responsibility of the authors and does not necessarily represent the official views of the National Institute of Arthritis and Musculoskeletal and Skin Diseases nor the National Institutes of Health. The funders had no role in study design, data collection and analysis, decision to publish, or preparation of the manuscript.

**Competing interests:** The authors have declared that no competing interests exist.

after surgery (10/10) demonstrating changes in the number of modules and/or organization of at least 1 module.

## Conclusions

This pilot work suggests that functional improvements following TKA may initially present as increases in the number of modules recruited during gait. Subsequent improvements in function may present as improved module organization.

## Noteworthy

This work is the first to characterize motor modules in TKA both before and after surgery and to demonstrate changes in the number and organization of modules over the time course of recovery, which may be related to changes in patient function. The plasticity of modular control following TKA is a key finding which has not been previously documented and may be useful in predicting or improving surgical outcomes through novel rehabilitation protocols.

## Introduction

Total knee arthroplasty (TKA) is the definitive end-stage treatment for knee osteoarthritis (KOA), with over 750,000 primary TKAs occurring in 2014 and over 1.25 million primary TKAs expected in the U.S. annually by 2030 [1]. While TKA improves function in most patients, more than 20% of adults demonstrate functional deficits 2-years post-TKA [2]. These functional deficits include slower walking speeds [3], stiff-knee gait [4], and difficulty in stair descent [5] and sit-to-stand transfer [6]. Several studies have suggested that these deficits are influenced by a variety of factors including implant design [7], surgical technique [8], ligament laxity and soft tissue balance [9], quadriceps strength [10], strength of the non-operated limb [11], and preoperative conditions [12]. However, the primary factor or combination of factors which reliably predict patient outcomes has yet to be determined.

Several studies have also examined neuromuscular control as a factor which may explain the functional deficits seen in this population. One study found that TKA patients, both before and 1-month after surgery, demonstrated higher co-activation of the quadriceps and hamstrings during stand-to-sit transfer and performed the five-times-sit-to-stand task more slowly than healthy adults [13]. Another study determined that TKA patients demonstrated quadriceps muscle activation deficits after surgery, but these deficits were not associated with quadriceps strength 3-months after TKA [14]. Some work has also been done to investigate changes in patient function and neural control in this population from the preoperative to postoperative condition. Hubley-Kozey et al. [15] studied patient function and muscle activation patterns in patients before and 1-year following TKA and found that while functional improvements were accompanied by shifts in activation patterns toward the patterns of asymptomatic individuals, the patterns were still statistically different from the asymptomatic individuals. While these studies provide valuable insight into the influence of activation patterns on function in KOA and TKA, it is still unknown how these electromyography (EMG)-measured factors may change over the time course of recovery after surgery. Further, it remains unclear if changes in co-activations and muscle activation patterns are indicative of changes in the underlying neuromuscular control strategies. Since individual EMG waveforms alone do not provide insight into the underlying mechanisms of control, other tools are necessary in order to determine the influence of neuromuscular control on patient function.

The theory of modular muscle control has gained popularity as a tool to explore neuromuscular control [16]. The theory describes a simplified neural control strategy in which a group of muscles is activated synergistically by a common neural command, reducing the set of activation profiles used to complete a motion [17]. The number of these commands, or "motor primitives" [18], and the organization of muscles within the modules, which are selected out of an individual's library of possible commands in order to complete a task, represent varying complexity of control patterns [19–21]. A higher number of modules used to complete a task, such as walking, may indicate a more complex neuromuscular control strategy and greater flexibility in control [21]. Muscles that are organized into a specific module activate as a group according to the module's timing profile, and recent literature has suggested that populations with neurologic movement disorders, including Parkinson's disease, post-stroke hemiparesis, and incomplete spinal cord injury, demonstrate altered module organization in comparison to healthy controls [22–25]. Recent literature has also indicated differences in motor modules between healthy adults and individuals with a variety of musculoskeletal knee conditions, including patellofemoral pain syndrome [26], anterior cruciate ligament deficiency [27], and KOA [28,29]. To our knowledge, there has been only one study examining modular control in the TKA population. Ardestani et al. [21] found that 1-year following a cruciate retaining (CR) TKA, a low-functioning group of participants demonstrated 2–3 motor modules while a high-functioning group demonstrated 4–5 modules, which more closely resembled a healthy control group that demonstrated 5–6 modules. The results of that study suggest that the complexity of neural control strategy may be an underlying mechanism that contributes to the large range of post-operative functional outcomes observed in this population. However, the relationship between module organization and function has yet to be characterized in the TKA population. Additionally, it has been suggested that rehabilitation programs may influence the complexity and organization of modular control in individuals with spinal cord injuries or neurological disorders [22,30]. However, it is still unknown whether changes in modular control exist in individuals with orthopaedic disorders and whether such changes may be related to the time course of functional improvements following surgical interventions, like TKA.

Therefore, the purpose of this pilot study was to determine the relationship between patient function and modular control during gait before and at two timepoints after surgery and to investigate the plasticity of neuromuscular control in the KOA and TKA populations. We hypothesized that better patient function before surgery, 6-months after surgery, and 24-months after surgery would be associated with (I) a higher number of modules and (II) module organization more similar to that of healthy controls. Additionally, we aimed to examine the relationship between changes in function and changes in the number and organization of modules over the time course of recovery from TKA.

## Materials and methods

### Data collection

Prior to enrolling in the original longitudinal study [31], 36 individuals (38 knees) with medial compartment KOA (19/19 R/L) provided written informed consent. All study procedures were approved by The Ohio State University Institutional Review Board. The cohort in the current study is a superset of that reported in Freisinger et al. [31], which examined gait biomechanics before surgery, and Chaudhari et al. [32], which examined muscle strength before and after surgery. Three orthopaedic surgeons (JFG, AHG, MDB; see acknowledgments) identified potential participants based on consultation for a TKA at The Ohio State Wexner Medical Center. Participants were required to have a body-mass index (BMI) less than 45, the ability to walk 20 meters without an assistive device, and no history of previous TKA or

osteotomy. Recruitment began in April 2012 and continued on a rolling basis, and data was collected from May 2012 to May 2017. All participants were scheduled to undergo a primary posterior-stabilizing (PS) TKA (Zimmer NexGen LPS Flex Knee) prior to study recruitment. By consensus, two fellowship-trained musculoskeletal radiologists (JP, AR; see acknowledgments) determined the Kellgren-Lawrence classification (KL-grade) of each participant's right and left knees [33].

The participants were tested in the Clinical, Functional, and Performance Biomechanics Laboratory at The Ohio State University approximately 1-month before, 6-months after, and 24-months after undergoing surgery. Each participant performed a minimum of 5 over-ground walking trials. Surface EMG data were collected at 1,500 Hz (Telemyo DTS System, Noraxon, Scottsdale, AZ) from 16 pre-gelled Ag/AgCl dual-electrodes (Model A10011, 10.592 mm sensor diameter, 40 mm inter-electrode distance; Vermed, Buffalo, NY) affixed over the bellies of 8 lower extremity muscles, bilaterally: rectus femoris, vastus lateralis, vastus medialis, biceps femoris, medial hamstrings, lateral gastrocnemius, medial gastrocnemius, and soleus. Prior to affixing the electrodes, the skin over each muscle was shaved to remove any hair and then cleansed and lightly abraded with alcohol wipes. Force plate data were collected at 1,500 Hz and were used to identify heel-strike-to-heel-strike gait cycle timing. Motion capture data were also collected but were not used in this study. Details of motion capture collection and analysis can be found in Freisinger et al. [31].

In addition to gait trials, several performance-based and patient-report measures of function were recorded. Each participant completed three clinical performance-based assessments: the timed stair-climbing test (SCT) [34], the timed up-and-go test (TUG) [35], and the six-minute walk test (6MW) [36]. Each participant also provided self-reported functional data using four of the Knee Injury and Osteoarthritis Outcome Score (KOOS) [37] survey sub-scales: pain, symptoms, activities of daily living (ADL), and quality of life (QOL). For each question, the participants' answers were scored on a scale from 0 to 4, with a higher score indicating better self-reported function. The scores were totaled within each subscale and normalized such that a maximum score of 100 represented high function and a minimum score of 0 represented poor function.

Due to attrition and technical challenges in the original longitudinal study, all participants were not represented at all three data collection time points. Two participants received a TKA on both left and right knees and participated in the study twice. Hence 36 participants enrolled, yet 38 knees were included in the study. High BMI (e.g. $33.9 \pm 5.1$ kg/m$^2$ before TKA), a common trait in individuals with osteoarthritis, caused soft tissue motion artifact in the EMG data in many of the trials. Due to this motion artifact, several trials were excluded from the study. Participants with fewer than 5 trials of usable EMG data at a particular testing time point were excluded from analysis at that time point. For these reasons and due to attrition, we were able to include data for 30 participants before surgery, 26 participants at 6-months post-TKA, and 13 participants at 24-months post-TKA (Table 1) in this pilot study. There were 8 participants with useable data at all three time points. All other participants had useable data at only two time points (11 before and 6-months after surgery, 2 before and 24-months after surgery, 2 at 6- and 24-months after surgery) or at only one time point (9 before surgery, 5 at 6-months after surgery, 1 at 24-months after surgery).

## Identification of muscle modules

We processed the EMG data for the TKA-involved limb from each gait trial to prepare for module extraction. The EMG data were demeaned and bandpass filtered to frequencies between 50 Hz and 300 Hz (Butterworth, 6th order) due to higher levels of motion artifact

**Table 1. Demographics, modules, and function.**

| | | Patient Population; n = 38 | | | | | Healthy Controls | Between Populations | |
|---|---|---|---|---|---|---|---|---|---|
| | | Pre-TKA | 6-months Post-TKA | 24-months Post-TKA | Between Time Points (GLMM) | | | (t-test or Chi-square test) | |
| | | | | | $p$ | F | | $p$ | $t/X^2$ |
| | | **n = 30** | **n = 26** | **n = 13** | | | **n = 10** | | |
| Demographics | Sex | 13M, 17F | 9M, 17F | 5M, 8F | - | - | 5M, 5F | - | - |
| | Height (m) | 1.70 ± 0.10 | 1.67 ± 0.10 | 1.69 ± 0.11 | 0.611 | $F_{2,66} = 0.50$ | 1.69 ± 0.08 | 0.906 | $t_{21} = -0.12$ |
| | Age (y) | 59.7 ± 7.8 | 60.3 ± 7.0 | 61.9 ± 6.9 | - | - | 63.5 ± 3.4 | 0.227 | $t_{21} = 1.24$ |
| | Mass (kg) | 96.3 ± 18.7 | 96.8 ± 20.3 | 100.5 ± 20.4 | 0.124 | $F_{2,13} = 2.46$ | 71.5 ± 13.5 | **0.001** | **$t_{21} = -3.72$** |
| | BMI (kg/m$^2$) | 33.7 ± 5.1 | 34.0 ± 5.7 | 35.5 ± 6.4 | **0.043**[bc] | **$F_{2,12} = 4.04$** | 24.8 ± 2.6 | **<0.001** | **$t_{21} = -4.56$** |
| | Walking Speed (m/s) | 0.97 ± 0.26 | 1.11 ± 0.22 | 1.22 ± 0.13 | **0.001**[ab] | **$F_{2,24} = 9.34$** | 1.12 ± 0.18 | 0.145 | $t_{21} = -1.51$ |
| | Normalized Stride Length (m/m) | 1.42 ± 0.25 | 1.53 ± 0.24 | 1.57 ± 0.22 | 0.056 | $F_{2,12} = 3.36$ | 1.51 ± 0.16 | 0.248 | $t_{21} = -1.19$ |
| Mods | Number of Modules | 2.5 ± 0.6 | 2.5 ± 0.6 | 2.5 ± 0.5 | 0.844 | $F_{2,18} = 0.17$ | 2.7 ± 0.5 | 0.673 | $X^2_1 = 0.178$ |
| | Proper Module Organization | 6 of 30 | 11 of 26 | 6 of 13 | 0.143 | $F_{2,19} = 2.15$ | - | - | - |
| Function Metrics | SCT (s) | 25.7 ± 14.6 | 19.5 ± 8.8[†] | 18.3 ± 6.8 | **0.049**[ab] | **$F_{2,16} = 3.65$** | - | - | - |
| | TUG (s) | 11.5 ± 4.4 | 10.0 ± 1.9 | 8.8 ± 1.8 | **0.016**[ab] | **$F_{2,15} = 5.46$** | - | - | - |
| | 6MW (m) | 401.1 ± 106.9 | 457.9 ± 89.0 | 495.5 ± 78.2 | **0.001**[abc] | **$F_{2,17} = 11.28$** | - | - | - |
| | KOOS-Pain (pts) | 48.7 ± 20.4 | 73.7 ± 22.2[†] | 79.6 ± 15.5 | **<0.001**[ab] | **$F_{2,14} = 21.99$** | - | - | - |
| | KOOS-Symptoms (pts) | 46.4 ± 21.2 | 63.6 ± 18.7[†] | 72.3 ± 15.9 | **0.006**[ab] | **$F_{2,12} = 7.91$** | - | - | - |
| | KOOS-ADL (pts) | 55.8 ± 20.8 | 78.2 ± 21.4[†] | 80.2 ± 14.1 | **<0.001**[ab] | **$F_{2,14} = 18.61$** | - | - | - |
| | KOOS-QOL (pts) | 24.7 ± 21.0 | 54.3 ± 22.2[†] | 64.1 ± 24.2 | **<0.001**[ab] | **$F_{2,10} = 27.51$** | - | - | - |

Average demographics, module metrics, and function metrics grouped by time point and population (± 1 standard deviation). The healthy control group was compared to the group of participants tested at 24-months post-TKA.

Symbols indicate differences between:

[a]Pre-TKA and 6-months Post-TKA

[b]Pre-TKA and 24-months Post-TKA

[c]6- and 24-months Post-TKA.

[†] Clinically meaningful improvement in function compared to the previous time point.

M = male.

F = female.

associated with high BMI values in this population, as recommended by Kieliba et al. [38] and Santuz et al. [39]. The EMG data were then full-wave rectified and smoothed using a 6 Hz low-pass filter (Butterworth, 6$^{th}$ order). All trials were examined visually, and those with missing channels, gaps in EMG data, or excessive motion artifact were excluded. We extracted the maximum number of available gait cycles from each participant, and those with fewer than 5 available gait cycles were excluded. Linear envelopes were formed by discretizing the data such

that each trial contained 201 data points, such that each point corresponds to 0.5% of the gait cycle, concatenating all available gait cycles, and normalizing each muscle first to the maximum value across all gait cycles and then to unit variance [40].

Muscle modules were calculated using non-negative matrix factorization (NMF) [41]. Given that neuron firing rates and strengths cannot be negative, NMF provides insight into an individual's neural control strategy that can be interpreted physiologically [16]. First, the linear envelopes for each subject were organized into $m \times t$ matrices representing the original EMG (EMG$_o$), where $m$ is the number of muscles (i.e. $m = 8$) and $t$ is the number of data points (i.e. $t = 201 \times$ the number of gait cycles available). The NMFs were performed using the Statistics and Machine Learning Toolbox in Matlab R2017a (The MathWorks Inc.; Natick, MA) which populated an $n \times t$ matrix (Pattern Matrix) and an $m \times n$ matrix (Weighting Matrix), where $n$ is the number of modules. The Pattern Matrix represents the temporal waveform of each module while the Weighting Matrix represents the weight of each muscle within each of the modules, such that the product of these matrices (EMG$_r$) reconstructs EMG$_o$ (Fig 1). The sum of

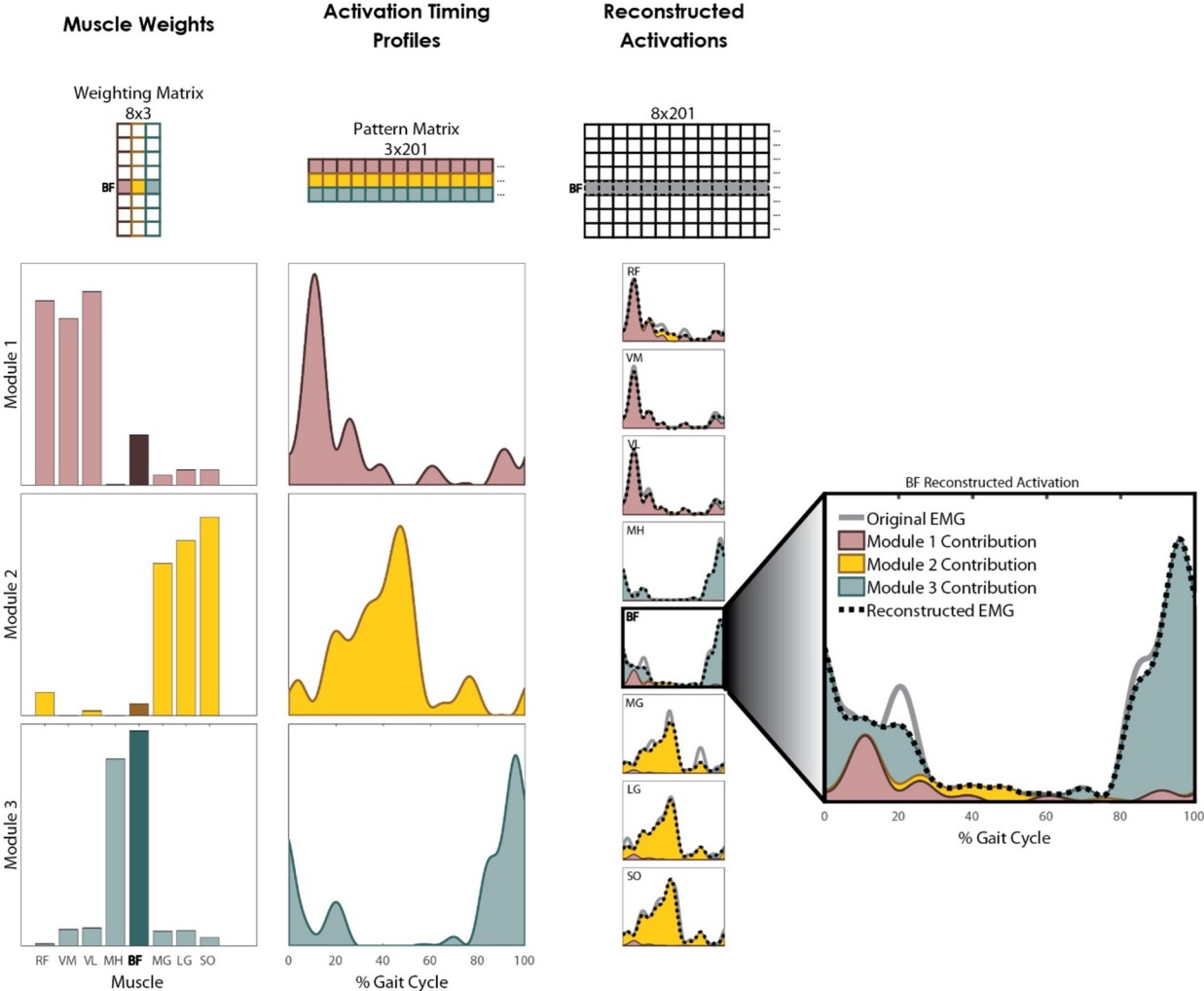

**Fig 1. Example of EMG reconstruction using non-negative matrix factorization.** Muscle modules were calculated using non-negative matrix factorization (NMF) in which an $m \times n$ Weighting Matrix is multiplied by an $n \times t$ Pattern Matrix to reconstruct the EMG patterns in an $m \times t$ matrix, where $m$ is the number of muscles (8 in this study), $n$ is the number of modules (3 in this example), and $t$ is the number of data points in the trial (201 in this study). The Pattern Matrix represents the timing profile of each module. The Weighting Matrix represents how strongly each muscle is represented in the module. This process is iterated, increasing $n$ from 1 to $m$ until the error between the reconstructed EMG (black dotted line) and the original EMG (solid gray line) is reduced to some acceptable level determined by the variability accounted for (VAF) criteria.

the squared errors between $EMG_r$ and $EMG_o$ was quantified with the percent of variability accounted for (VAF). The VAF was calculated for each muscle (mVAF) of each solution using:

$$mVAF = \left( 1 - \frac{\left(EMG_{o,m} - EMG_{r,m}\right)^2}{EMG_{o,m}^2} \right) \times 100\% \qquad \text{Eq 1}$$

and the total VAF (tVAF) for each solution was calculated using:

$$tVAF = \left( 1 - \frac{\sum_{m=1}^{8} \left(EMG_{o,m} - EMG_{r,m}\right)^2}{\sum_{m=1}^{8} EMG_{o,m}^2} \right) \times 100\% \qquad \text{Eq 2}$$

In order to find the number of modules required to reconstruct $EMG_o$, the number of modules was increased from 1 to *m* (i.e., *m* = 8; the number of muscles recorded) in separate NMFs with a maximum of 10,000 iterations until the following criteria were satisfied: the lower bound of the 95% confidence interval (CI) of tVAF was greater than 90% for all muscles and (a) the minimum lower bound of the 95% CI of all mVAFs was greater than 75%, or (b) the minimum lower bound of the 95% CI for all mVAFs was greater than 75% and the addition of another module did not raise the lower bound of the 95% CI for that mVAF by more than 5% of its value in the previous NMF [24]. The 95% CIs of the VAFs were found using a bootstrapping technique adopted from Allen et al. [22], wherein $EMG_o$ was resampled 500 times with replacement. The 95% CI was assembled from the VAFs of all 500 bootstrapped samples. Once the appropriate number of modules was determined, we ran a final NMF on the original EMG using a multiplicative update algorithm with 50 replicates. To allow for comparison of modules between participants and timepoints, the Weighting Matrix was normalized to the maximum muscle weight within that module. Additionally, we evaluated tVAF for all possible module solutions (i.e. 1 module, 2 modules, . . ., 8 modules) as a measure of the complexity of the control strategy. A higher tVAF value for any given module solution represents a better fit between $EMG_r$ and $EMG_o$. Therefore, a low tVAF value indicates that the solution is not capable of fully capturing the complexity of the actual control strategy.

To characterize modular organization, we adapted definitions of organization characteristics, $W_{musc}$ and $W_{sum}$, from Hayes et al. [25]. $W_{musc}$ is defined as the number of significantly active muscles in each module, and $W_{sum}$ is defined as the sum of the weights of the significantly active muscles in each module. In Hayes et al. [25] a muscle was considered significantly active if the 95% CI of weights for the 500 bootstrapped gait cycles did not contain 0. However, participants in the current study often had confidence intervals with a lower bound greater than zero but a very small upper bound. Under the original definition from Hayes et al. [25], all modules for all participants in the TKA cohort had a $W_{musc}$ of 7 or 8, meaning all muscles were significantly active in all modules. To uncover more subtle differences in $W_{musc}$ and $W_{sum}$ between participants in our cohort, we adapted the definition of "significantly active," such that a muscle was considered significantly active if the 95% CI of weights did not contain 0 and the upper bound was greater than 0.25. Modules from each of the 500 samples were sorted using a k-means algorithm to ensure like-modules were grouped prior to determining the 95% CIs for the weights [42]. The significantly active muscles in each module were counted ($W_{musc}$) and the weights of these muscles were added ($W_{sum}$).

### Healthy control group

Motor modules from 10 older adults with no known lower extremity osteoarthritis, a subset of the cohort used in Roelker et al. [42], were used as healthy, age-matched controls to compare

to the KOA and TKA data. The processes for data collection and identification of muscle modules are consistent with this study, with the exception of the bandpass filter used to process the EMG data. Roelker et al. [42] used a 30–300 Hz bandpass filter while we used a 50–300 Hz bandpass filter in the current study due to the previously mentioned high BMI and resulting motion artifact observed in the TKA cohort. We determined $W_{musc}$ and $W_{sum}$ for each module for each healthy participant based on our new definition for "significantly active" muscles.

## Comparison of module organization

To determine similarities in module organization between populations, we compared the module weightings of the study participants to those of the healthy controls. The muscle weights of each module for each participant at each timepoint were compared to the averaged weights of healthy age-matched controls with the same number of modules using Pearson correlations. A critical correlation coefficient of $\rho \geq 0.834$ was selected based on the number of muscles (8) and a $p$-value of 0.01 [22,23,25]. A module was considered to have proper organization if its weights were correlated with the healthy group and its $W_{musc}$ and $W_{sum}$ values were within the 95% CIs of the healthy group. Participants were considered to have proper organization if all of their modules were organized like healthy controls and poor organization if at least one module did not have healthy organization.

## Clinically meaningful changes in functional performance

Clinically meaningful differences in performance-based and self-reported measures were evaluated based on minimum detectable changes (MDCs), as defined by previously reported 90% CIs from representative populations. Improvements were indicated by decreases in the time to complete SCT ($\geq 1.9$ seconds; [43]) and TUG ($\geq 2.49$ seconds; [36]), increases in the distance walked during 6MW ($\geq 61.34$ meters; [36]), and increases in all KOOS subscales ($\geq 10$ points; [37]).

## Statistics

To address our hypotheses, we investigated the relationship between functional performance measures and modules in terms of their number (Hypothesis I) and organization (Hypothesis II). At each time point, the functional measures were summarized by the number or organization of modules. Generalized linear mixed models (GLMM) for repeated measures were used to estimate the effects of number of modules and module organization (proper or poor) on all functional measures at each time point. Time point, module number/organization, and their interaction were included as fixed effects and participants were included as random effects in the GLMMs. GLMMs allowed us to include all participants, even with missing data, assuming data was missing at random in the analysis.

We examined differences in demographics, function, and modules between time points for the TKA cohort and also examined differences in demographics and number of modules between the TKA and healthy cohorts. Separate GLMMs were used to estimate the changes over time in demographics data (height, mass, BMI, self-selected walking speed, stride length normalized to leg length), performance-based function (SCT, TUG, 6MW), KOOS-subscale scores (Pain, Symptoms, ADL, QOL), number of modules, and module organization (proper or poor). Separate unpaired, two-tailed t-tests (all data was normally distributed; Anderson-Darling test for normality, all $p \geq 0.052$) compared demographics data between the TKA group at 24-months after surgery and the healthy control group. A Chi-square test compared number of modules between the healthy control group and the group tested at 24-months

post-TKA. As an additional examination of module complexity, we used GLMMs to estimate differences in tVAF for all possible module solutions (i.e. 1 to 8 modules) between time points.

All GLMMs were performed using Statistical Analysis Software (SAS Institute Inc.; Cary, NC). Main effects were estimated from the fixed effects parameters or their corresponding contract of the parameters from the GLMM, and, if significant, pairwise differences were assessed based on the least squares means difference. All other statistical analysis was performed using Minitab (Minitab, LLC; State College, PA). A significance level of $\alpha < 0.05$ was established *a priori* for all tests.

## Results

### Population demographics and modules

The TKA group had statistically significant changes over time in self-selected walking speed ($p = 0.001$) and BMI ($p = 0.043$; Table 1). Self-selected walking speed increased over time and was slower before surgery than at 6- (pairwise $p = 0.007$) and 24-months (pairwise $p < 0.001$) after surgery. BMI also increased over time and was higher BMI 24-months after surgery than before (pairwise $p = 0.018$) or 6-months (pairwise $p = 0.049$) after surgery. Between the TKA cohort (24-months after surgery) and the healthy age-matched cohort, there were no significant differences in height, self-selected walking speed, or normalized stride length (all $p \geq 0.145$), but the TKA group had significantly higher mass ($p = 0.001$) and BMI ($p < 0.001$) than the healthy group (Table 1).

Of the 10 healthy individuals, 3 demonstrated 2 modules and 7 demonstrated 3 modules ($2.7 \pm 0.5$ modules; Table 1). Those with 2 modules had 1 module dominated by the plantarflexors and 1 module dominated by the quadriceps and hamstrings. Those with 3 modules had modules dominated by the plantarflexors, quadriceps, and hamstrings, separately. Before surgery, 16 individuals with TKA demonstrated 2 modules and 14 demonstrated 3 modules, and 6 of the 30 (20.0%) participants had proper organization. By 6-months after surgery, 12 demonstrated 2 modules and 14 demonstrated 3 modules, and 11 of the 26 (42.3%) participants had proper module organization. By 24-months after surgery, 5 participants demonstrated 2 modules and 8 demonstrated 3 modules, and 6 of the 13 (46.2%) participants had proper organization. There was no difference in number of modules ($p = 0.844$) or module organization ($p = 0.143$) between time points (Table 1).

There were differences in mean tVAF values between time points for solutions with 1, 2, 6, and 7 modules (Fig 2). For the 1-module solution, the group tested before surgery ($79.8 \pm 6.3\%$) had a statistically higher mean tVAF value than those at 6-months ($75.7 \pm 6.3\%$; pairwise $p = 0.001$) and 24-months ($72.9 \pm 8.2\%$; pairwise $p = 0.003$) after surgery. For the 2-module solution, the group tested before surgery ($91.2 \pm 2.6\%$) had a statistically higher mean tVAF than those at 24-months after surgery ($89.3 \pm 2.7\%$; pairwise $p = 0.033$). For the 6-module solution, the group tested before surgery ($99.3 \pm 0.3\%$) had a statistically higher mean tVAF than those at 24-months after surgery ($99.1 \pm 0.3\%$; pairwise $p = 0.016$). For the 7-module solution, the group tested before surgery ($99.7 \pm 0.1\%$) had a statistically higher mean tVAF than those at 6-months ($99.7 \pm 0.1\%$; pairwise $p = 0.040$) and 24-months ($99.6 \pm 0.2\%$; pairwise $p = 0.004$) after surgery.

The worst averaged functional scores were observed before surgery and the best averaged functional scores were observed 24-months after surgery (Table 1). All measures were statistically different between timepoints (all $p \leq 0.049$). From before to 6-months after surgery, all measures had statistically significant improvements (all pairwise $p \leq 0.031$), and these improvements were also clinically meaningful for the SCT and all KOOS subscales. From before to 24-months after surgery, all measures had statistically significant (all pairwise

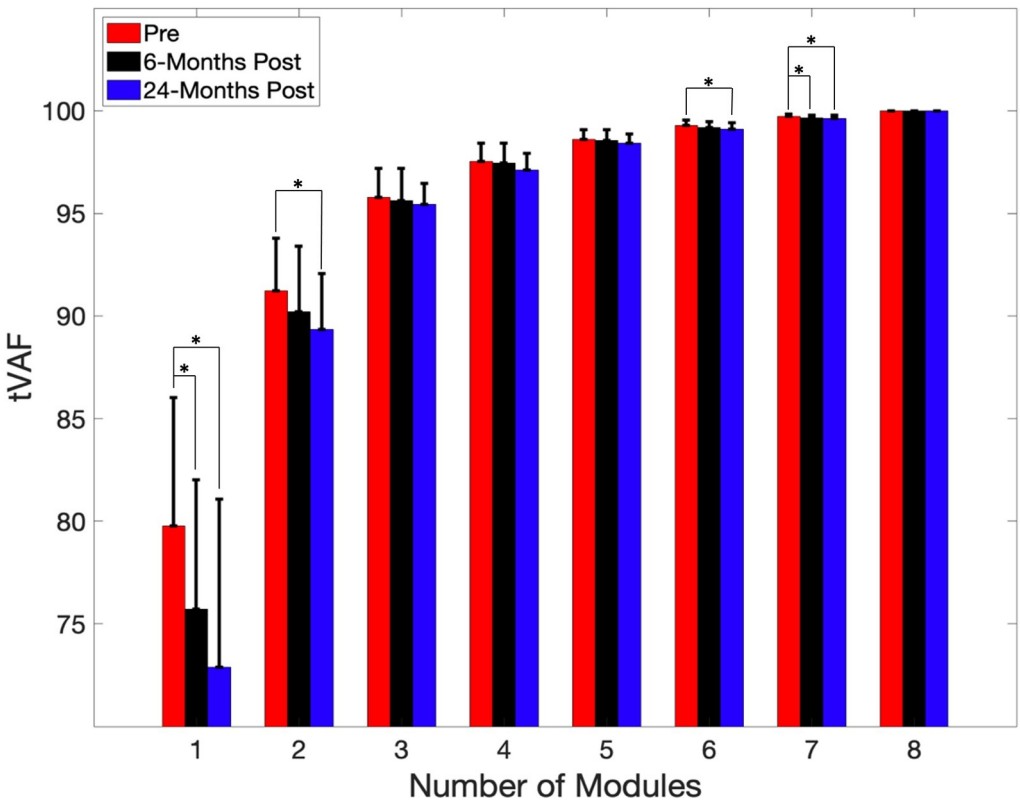

**Fig 2. tVAF for all module solutions.** Estimated mean tVAF by time point for number of modules varying from 1 to 8. Error bars represent +1 standard deviation. * indicates statistically significant differences between time points.

$p \leq 0.020$) and clinically meaningful improvements. Average 6MW distance also had a statistically significantly improvement from 6- to 24-months after surgery (pairwise $p = 0.036$), but this improvement was not clinically meaningful.

### Hypothesis I: Function and number of modules

A higher number of modules was associated with better performance in a few performance-based and patient-reported functional measures before and 6-months after surgery but with poorer performance in a few functional measures 24-months after surgery (Figs 3 & 4). Before surgery, participants with 3 modules demonstrated better scores on average than those with 2 modules in all measures, and this difference was clinically meaningful in all measures except the 6MW distance (Table 2). These results were statistically significant for the TUG ($p = 0.002$) and KOOS-Pain ($p = 0.005$) scores and approached significance for the SCT ($p = 0.060$), KOOS-ADL ($p = 0.057$), and KOOS-QOL ($p = 0.065$) scores. By 6-months after surgery, participants with 3 modules demonstrated better scores on average than those with 2 modules in all measures except the KOOS-QOL subscale (Table 2). These differences were clinically meaningful for only the SCT, which also approached statistical significance ($p = 0.052$) and were statistically significant but not clinically meaningful for the TUG test ($p = 0.028$). By 24-months after surgery, participants with 2 modules had better scores on average than those with 3 modules in SCT, 6MW, KOOS-Symptoms scores (Table 2). These differences were clinically meaningful for the SCT and for the 6MW, which was also statistically significant ($p = 0.006$).

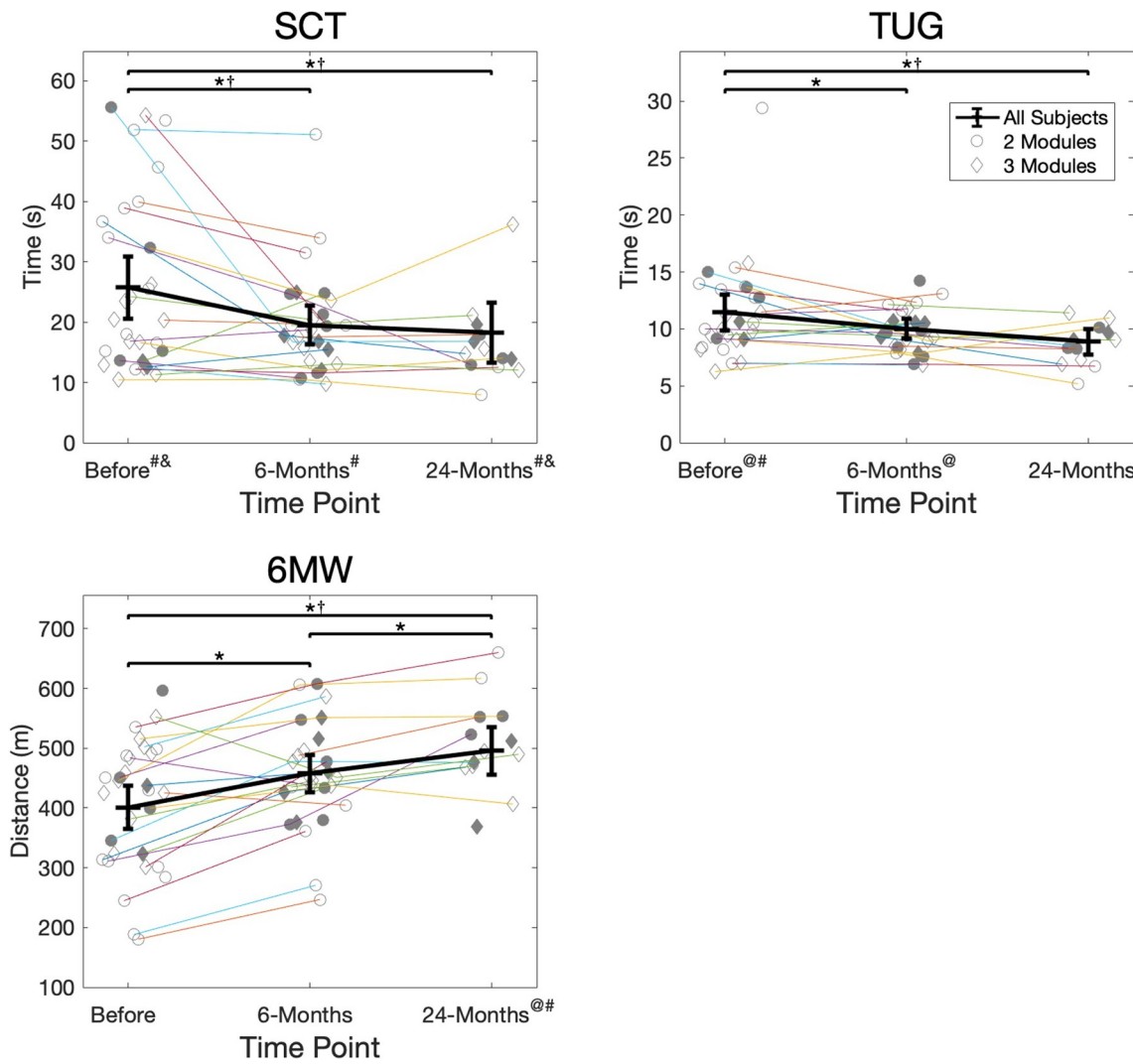

**Fig 3. Performance-based function.** Performance-based functional scores (SCT, TUG, 6MW) for each participant at each time point. Participants who were present for consecutive testing time points are connected. Filled symbols indicate participants with proper module organization. Thick black lines indicate the group estimated means and 95% confidence intervals at each time point, determined by the GLMM. */† = statistically significant (*) and clinically meaningful (†) differences in function between time points. @/# = statistically significant (@) and clinically meaningful (#) differences in function between groups demonstrating the same number of modules at each time point. & = clinically meaningful differences in function between groups demonstrating proper or poor module organization at each time point. There were no statistically significant differences.

### Hypothesis II: Function and module organization

Better module organization was not statistically associated with better performance-based function (SCT, TUG, or 6MW) or patient-reported function (KOOS subscales) before or after surgery (Figs 3 & 4). However, the largest differences in function between the proper and poor organization groups occurred 24-months after surgery, with the proper organization group demonstrating better function on average in all performance-based and patient-reported measures (Table 3). These results approached statistical significance for the KOOS-Pain subscale ($p = 0.061$) and were clinically meaningful for the SCT, KOOS-Pain, and KOOS-Symptoms scores.

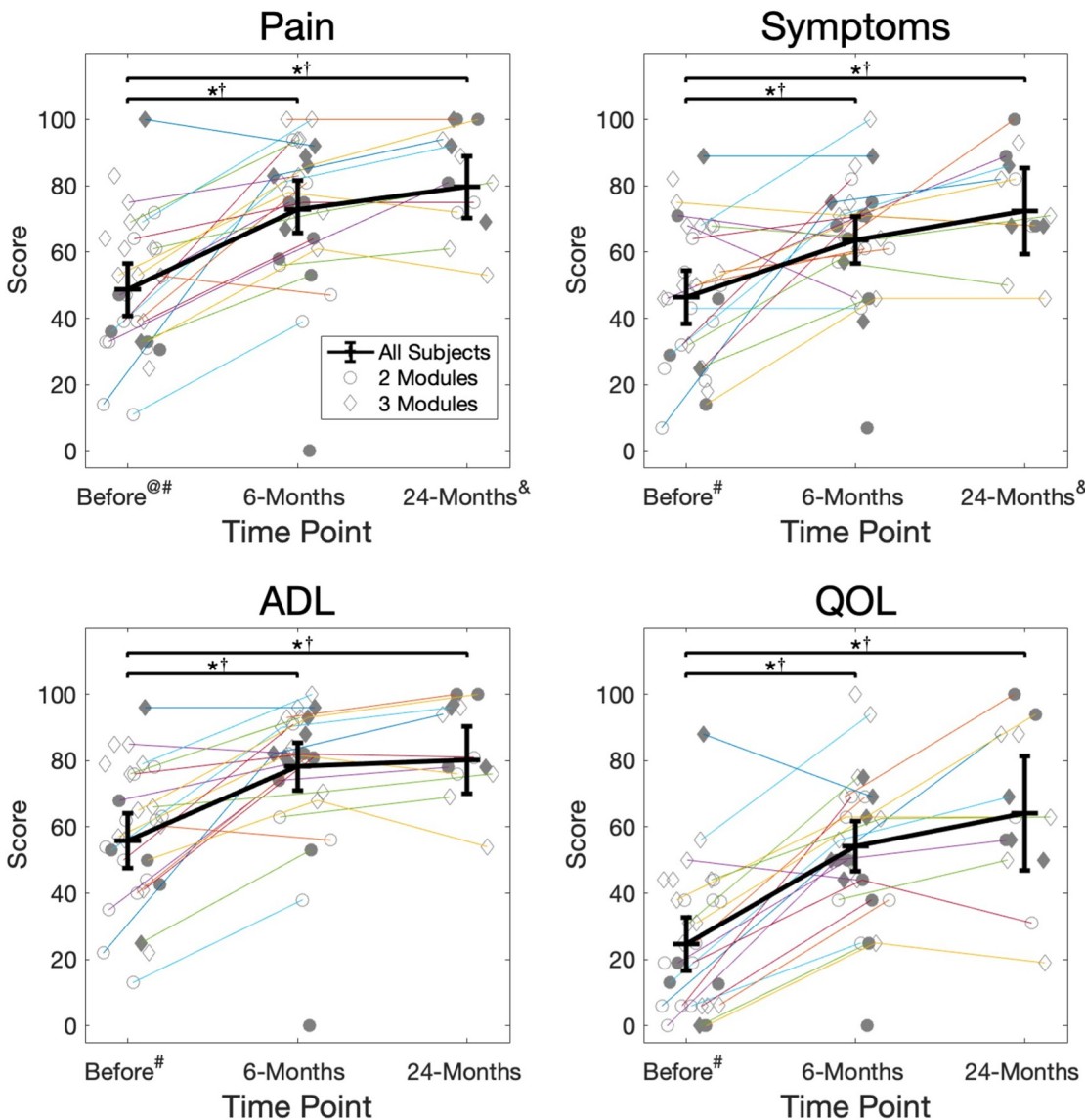

**Fig 4. Patient-reported function.** KOOS survey subscale scores (Pain, Symptoms, ADL, QOL) for each participant at each time point. Participants who were present for consecutive testing time points are connected. Filled symbols indicate participants with proper module organization. Thick black lines indicate the group estimated means and 95% confidence intervals at each time point, determined by the GLMM. */† = statistically significant (*) and clinically meaningful (†) differences in function between time points. @/# = statistically significant (@) and clinically meaningful (#) differences in function between groups demonstrating the same number of modules at each time point. & = clinically meaningful differences in function between groups demonstrating proper or poor module organization at each time point. There were no statistically significant differences.

## Secondary analysis: Changes in modules

While there were no significant differences in modules between participants grouped by time point, we observed changes in module number and organization in individual participants throughout the time course of recovery (Table 4). From before to 6-months after surgery, 63.2% of participants present at both time points (12/19) demonstrated changes in module number and/or organization. From 6- to 24-months after surgery, 80.0% of participants present at both time points (8/10) demonstrated changes in module number and/or organization. From before to 24-months after surgery, 100% of participants present at both time points (10/

**Table 2. Functional measures within module number groups.**

| | | Two Modules | Three Modules | *p* | Estimate | 95% CI |
|---|---|---|---|---|---|---|
| Number of Subjects | Pre-TKA | n = 16 | n = 14 | - | - | - |
| | 6-months Post-TKA | n = 12 | n = 14 | - | - | - |
| | 24-months Post-TKA | n = 5 | n = 8 | - | - | - |
| SCT (s) | Pre-TKA | 30.2 ± 15.3 | 21.0 ± 11.2 | 0.060[†] | 9.2 | [-0.42, 18.90] |
| | 6-months Post-TKA | 22.5 ± 11.5 | 16.4 ± 4.2 | 0.052[†] | 6.1 | [-0.05, 12.19] |
| | 24-months Post-TKA | 16.1 ± 3.6 | 19.5 ± 7.6 | 0.434[†] | -3.4 | [-12.44, 5.70] |
| TUG (s) | Pre-TKA | 13.1 ± 5.3 | 9.6 ± 2.3 | **0.002[†]** | **3.5** | **[1.63, 5.35]** |
| | 6-months Post-TKA | 10.8 ± 2.5 | 9.4 ± 1.4 | **0.028** | **1.4** | **[0.17, 2.60]** |
| | 24-months Post-TKA | 9.2 ± 1.9 | 8.9 ± 1.6 | 0.803 | 0.3 | [-2.51, 3.16] |
| 6MW (m) | Pre-TKA | 379.7 ± 123.8 | 425.9 ± 76.8 | 0.128 | -46.7 | [-107.79, 14.49] |
| | 6-months Post-TKA | 453.8 ± 116.3 | 467.9 ± 53.7 | 0.621 | -13.1 | [-67.74, 41.44] |
| | 24-months Post-TKA | 546.1 ± 55.7 | 468.4 ± 48.4 | **0.006[†]** | **78.5** | **[29.27, 127.72]** |
| KOOS-Pain (points) | Pre-TKA | 39.2 ± 16.6 | 58.8 ± 19.8 | **0.005[†]** | **-19.6** | **[-32.58, -6.57]** |
| | 6-months Post-TKA | 69.8 ± 24.6 | 78.3 ± 12.4 | 0.186 | -8.6 | [-21.82, 4.67] |
| | 24-months Post-TKA | 80.6 ± 13.5 | 80.7 ± 17.0 | 0.987 | -0.1 | [-19.98, 19.68] |
| KOOS-Symptom (points) | Pre-TKA | 40.3 ± 18.8 | 52.4 ± 22.5 | 0.152[†] | -12.1 | [-28.94, 4.72] |
| | 6-months Post-TKA | 60.5 ± 19.8 | 67.1 ± 17.3 | 0.393 | -6.6 | [-22.92, 9.71] |
| | 24-months Post-TKA | 79.0 ± 13.8 | 69.2 ± 16.5 | 0.418 | 9.7 | [-15.79, 35.26] |
| KOOS-ADL (points) | Pre-TKA | 49.1 ± 17.9 | 63.0 ± 22.3 | 0.057[†] | -13.9 | [-28.17, 0.46] |
| | 6-months Post-TKA | 76.4 ± 25.6 | 80.0 ± 9.5 | 0.482 | -3.6 | [-15.35, 8.08] |
| | 24-months Post-TKA | 77.8 ± 12.0 | 83.2 ± 15.9 | 0.520 | -5.4 | [-23.85, 12.98] |
| KOOS-QOL (points) | Pre-TKA | 17.7 ± 15.0 | 32.3 ± 24.1 | 0.065[†] | -14.7 | [-30.49, 1.16] |
| | 6-months Post-TKA | 57.2 ± 20.0 | 53.1 ± 20.1 | 0.515 | 4.1 | [-9.36, 17.55] |
| | 24-months Post-TKA | 61.7 ± 28.4 | 64.8 ± 22.5 | 0.832 | -3.1 | [-34.98, 28.69] |

Average functional measures within groups of participants demonstrating the same number of modules at each timepoint (estimated mean ± 1 standard deviation).

[†] Clinically meaningful difference in function between the Two Modules and Three Modules groups.

10) demonstrated changes in module number and/or the organization of at least 1 module (Table 4).

## Discussion

In an effort to identify neuromuscular control strategy as a factor related to functional deficits in individuals after TKA, the purpose of this study was to determine the relationship between modular control strategy and function in individuals before and at two time points after TKA. Our findings partially confirmed our first hypothesis that better function would be related to a higher number of modules. We found that participants with better performance-based function before and 6-months after surgery and better patient-reported function before surgery had a higher number of modules. However, at 24-months after surgery, a higher number of modules were related to worse 6MW performance, which contradicts our first hypothesis. Our findings did not strongly confirm our second hypothesis that better function would be related to module organization that more closely resembled that of healthy controls. However, by 24-months after surgery, participants with proper module organization had better function on average than those with poor organization in all performance-based and patient-reported measures, with some differences being clinically meaningful. Considerable participant drop-out by the 24-month time point may have prevented us from detecting statistically significant

**Table 3. Functional measures within module organization groups.**

| | | Poor | Proper | p-Value | Estimate | 95% CI |
|---|---|---|---|---|---|---|
| Number of Subjects | Pre-TKA | n = 24 | n = 6 | - | - | - |
| | 6-months Post-TKA | n = 15 | n = 11 | - | - | - |
| | 24-months Post-TKA | n = 7 | n = 6 | - | - | - |
| SCT (s) | Pre-TKA | 26.1 ± 14.2 | 24.2 ± 17.3 | 0.750[†] | 2.0 | [-10.64, 14.60] |
| | 6-months Post-TKA | 19.9 ± 10.7 | 19.1 ± 5.4 | 0.824 | 0.7 | [-6.39, 7.88] |
| | 24-months Post-TKA | 19.6 ± 9.3 | 16.2 ± 2.6 | 0.487[†] | 3.4 | [-7.06, 13.93] |
| TUG (s) | Pre-TKA | 11.3 ± 4.8 | 12.0 ± 2.5 | 0.695 | -0.6 | [-4.04, 2.78] |
| | 6-months Post-TKA | 10.2 ± 1.9 | 9.8 ± 2.0 | 0.679 | 0.3 | [-1.25, 1.88] |
| | 24-months Post-TKA | 8.5 ± 2.3 | 9.2 ± 0.8 | 0.540 | -0.7 | [-3.11, 1.72] |
| 6MW (m) | Pre-TKA | 401.6 ± 110.3 | 399.9 ± 97.4 | 0.963 | 1.7 | [-75.03, 78.46] |
| | 6-months Post-TKA | 454.4 ± 96.2 | 462.2 ± 79.6 | 0.762 | -7.7 | [-60.33, 44.85] |
| | 24-months Post-TKA | 489.3 ± 89.8 | 502.0 ± 69.3 | 0.727 | -12.7 | [-92.51, 67.11] |
| KOOS-Pain (points) | Pre-TKA | 49.5 ± 19.2 | 46.4 ± 26.8 | 0.729 | 3.2 | [-15.41, 21.74] |
| | 6-months Post-TKA | 77.3 ± 19.1 | 68.1 ± 25.7 | 0.116 | 9.2 | [-2.75, 21.19] |
| | 24-months Post-TKA | 75.2 ± 14.6 | 86.1 ± 12.8 | 0.061[†] | -11.0 | [-22.53, 0.60] |
| KOOS-Symptom (points) | Pre-TKA | 46.8 ± 19.6 | 45.7 ± 29.1 | 0.920 | 1.1 | [-20.41, 22.54] |
| | 6-months Post-TKA | 63.4 ± 15.7 | 63.4 ± 22.5 | 0.999 | 0.0 | [-18.11, 18.09] |
| | 24-months Post-TKA | 67.3 ± 17.3 | 79.1 ± 13.8 | 0.315[†] | -11.8 | [-36.93, 13.40] |
| KOOS-ADL (points) | Pre-TKA | 56.1 ± 20.4 | 55.9 ± 24.2 | 0.984 | 0.2 | [-18.51, 18.88] |
| | 6-months Post-TKA | 79.2 ± 17.1 | 76.6 ± 26.8 | 0.641 | 2.6 | [-10.72, 15.97] |
| | 24-months Post-TKA | 78.7 ± 14.5 | 83.9 ± 10.6 | 0.478 | -5.2 | [-21.13, 10.70] |
| KOOS-QOL (points) | Pre-TKA | 24.4 ± 17.7 | 27.5 ± 33.2 | 0.756 | -3.0 | [-23.10, 17.01] |
| | 6-months Post-TKA | 56.8 ± 22.4 | 50.5 ± 20.8 | 0.355 | 6.3 | [-9.43, 21.97] |
| | 24-months Post-TKA | 63.2 ± 26.3 | 67.7 ± 21.3 | 0.759 | -4.5 | [-36.12, 27.19] |

Average functional measures within groups of participants with modules that were (Proper) or were not (Poor) organized like healthy controls at each timepoint (estimated mean ± 1 standard deviation).

[†] Clinically meaningful difference in function between the Proper and Poor organization groups.

**Table 4. Count of participants with changes in modules.**

| | | Pre-TKA to 6-m Post-TKA | 6-m to 24-m Post-TKA | Pre-TKA to 24-m Post-TKA |
|---|---|---|---|---|
| | | n = 19 | n = 10 | n = 10 |
| Number of Modules | Increase | 3 (15.8%) | 1 (10.0%) | 4 (40.0%) |
| | No Change | 12 (63.2%) | 7 (70.0%) | 4 (40.0%) |
| | Decrease | 4 (21.1%) | 2 (20.0%) | 2 (20.0%) |
| Proper vs. Poor Organization | Improved | 5 (26.3%) | 2 (20.0%) | 4 (40.0%) |
| | No Change | 12 (63.2%) | 6 (60.0%) | 5 (50.0%) |
| | Worsened | 2 (10.5%) | 2 (20.0%) | 1 (10.0%) |
| Any Module Characteristic | Change | 12 (63.2%) | 8 (80.0%) | 10 (100.0%) |
| | No Change | 7 (36.8%) | 2 (20.0%) | 0 (0.0%) |

Number of participants (%) that were present at multiple time points and demonstrated changes in module characteristics. Proper organization describes participants whose organization matched those of healthy controls for all modules. Poor organization describes participants with at least 1 module that was not organized like healthy controls. Participants listed as having a change in "Any Module Characteristic" had a change in the number of modules and/or the organization of at least 1 module compared to healthy controls.

differences in function between groups demonstrating proper and poor module organization. While we did not find differences in population-averaged modular control between time-points, we did observe changes in both module number and organization in individuals and in population-mean complexity, as measured by tVAF. While the number of motor modules is often used as a measure of neuromuscular control complexity, complexity has also been quantified using tVAF values in previous literature [44–46]. Given that all participants demonstrated only 2 or 3 modules in the current study, tVAF provided greater resolution to investigate changes in control complexity over time. While the differences in tVAF between testing time points were statistically significant for the 6- and 7-module solutions, they were likely not clinically meaningful, though a clinically meaningful difference in tVAF values has not been established in the literature. Larger differences in tVAF between time points were observed for the 1- and 2-module solutions, which may indicate that module complexity is malleable in this population. Together with our observations on module organization, these results indicate that motor modules are plastic and suggest that neural control strategy is influenced by surgical intervention and rehabilitation in this population.

This study was motivated by previous work which has found that a higher number of modules is related to better function [21,24,25] and by our own previous work [29] where we observed decreases in the number of modules in healthy older adults and older adults with knee osteoarthritis (KOA) compared to healthy younger adults. In all of these studies, healthy older adults have been found to demonstrate a range in the number of motor modules, from as low as 2 to as high as 6, depending on the extraction method and EMG sensor pattern [21,24,25,29]. In the current study, we used the healthy subjects from Roelker et al. [29] for direct comparison of the number of modules between individuals with TKA and healthy adults using the same module extraction methods and experimental configuration. In Roelker et al. [29], we observed that healthy older adults demonstrated only 2 or 3 modules despite demonstrating similar self-selected walking speed to healthy younger adults, who demonstrated up to 4 modules. Given that there were no differences in self-selected walking speed between the healthy younger and healthy older adults, we believe this healthy older adult cohort was a suitable control group for the current study.

The association we observed between number of modules and function partially confirms the findings of Ardestani et al. [21], who found that a higher number of modules was associated with better patient-reported function 1-year following TKA. While we found that better function was associated with a higher number of modules in individuals with KOA prior to TKA, we did not find this association by 24-months after surgery. Several methodological differences exist between the current study and Ardestani et al. [21] which may explain the differences in results. Ardestani et al. [21] measured patient-reported function using the Knee Functional Survey (KFS) [47], which assesses general patient satisfaction and symptoms, ability to complete activities of daily living, and ability to participate in sports and exercise. The KFS may most closely relate to the KOOS-Symptoms, KOOS-ADL, and KOOS-QOL sub-scales, which were not statistically related to number of modules in the current study. However, our small sample sizes, particularly by 24-months after surgery, may have prevented us from detecting associations between patient-reported function and number of modules after surgery. Additionally, the methods for module extraction used by Ardestani et al. [21], including a strict error criterion of 90% accuracy between the experimental and reconstructed EMG, resulted in a range of 2–5 modules, whereas we saw only 2–3 modules. We believe our approach for calculating modules using 500 bootstrapped samples [48] and a previously developed, conservative VAF criteria [24] is robust. Lastly, participants in Ardestani et al. [21] received cruciate-retaining implants while participants in the current study received posterior-stabilizing implants. The comparison of our results with those of Ardestani et al. [21] may

suggest that proprioceptive inputs, which are lost as a result of sacrificing the cruciate ligaments, influence neuromuscular control strategy. However, the lack of consistency in methodologies across studies warrants further investigation into the influence of retaining the posterior cruciate ligament on post-surgical neuromuscular control. Further, while Ardestani et al. [21] indicated an association between a higher number of modules and better patient-reported function at 1-year following TKA, our results indicate that the relationship between number of modules and function changes over the time course of recovery.

The observed relationship between a higher number of modules and better function in some measures before TKA reversed in some measures by 24-months after TKA. The group with 2 modules 24-months after surgery demonstrated 6MW distances that were on average better, statistically and clinically, than those with 3 modules. Further analysis revealed that there were 2 participants with 2 modules that had significantly higher 6MW distances than the other participants at this time point which drove this result. Both participants demonstrated module organization that was different from the healthy controls, with one module that was dominated by the quadriceps and hamstrings and one module that was dominated by the plantarflexors and hamstrings such that the hamstrings were constantly active throughout the gait cycle. These participants also demonstrated better-than-average SCT and TUG performance at 24-months but had unremarkable KOOS scores and demographics measures. When examining all participants at 24-months post-TKA, we found that 3 of the 5 participants (60.0%) with 2 modules had proper organization while only 3 of the 8 participants (37.5%) with 3 modules had proper organization. These results indicate that module organization, rather than number, may be a factor which is related to function in the long-term post-operative time frame (24-months after TKA).

Though there were no statistically significant differences in function between the proper and poor organization groups at any time point, we observed trends toward better function in all functional measures, except TUG, for the group with proper module organization at 24-months after TKA. While the number of modules has frequently been used as a metric for motor control, recent studies suggest that module analysis which examines the composition of modules across multiple trials may provide further insight into functional performance in tasks like walking [22]. Furthermore, relationships between module organization and function have been observed in populations with neurologic movement disorders. Brough et al. [49] found that individuals with post-stroke hemiparesis without an independent plantarflexors module demonstrated poorer locomotor function compared to patients and controls with an independent plantarflexors module. Hayes et al. [25] also observed altered module composition and slower walking speeds in individuals with chronic incomplete spinal cord injury compared to able-bodied individuals. While there was not a significant relationship in the current study between function and module organization, as it relates to the organization of healthy individuals, the observed trends toward better function in those with module organization similar to healthy controls, with several measures presenting with clinically meaningful differences between groups, indicates that there may be a relationship between module composition and function which should be explored further.

Investigation of the number and organization of modules over time revealed that neuromuscular control was plastic in individual participants, and this neuroplasticity may be related to functional outcomes. Previously, Hubley-Kozey et al. [15] found that shifts in activation patterns of some muscles from before surgery to 1-year after TKA were accompanied by improvements in knee flexor and extensor strength and patient-reported outcomes (via WOMAC Index [50]) but were still statistically different from the patterns of asymptomatic individuals. In the present study, we observed statistically significant and clinically meaningful improvements in all functional measures from before to after surgery which accompanied population-mean decreases in several tVAF measures. While small sample sizes (n < 10 present at all time

points) prevented us from using statistical analysis to assess the association between changes in function and changes in module characteristics between time points, these results suggest that there may be an underlying relationship between improvements in function and increased module complexity. Further, when comparing function at 6- to 24-months, there was a larger improvement in all functional measures for groups with proper organization than those with poor organization, with the exception of TUG (Table 3). This observation may indicate that individuals with proper module organization may have a greater capacity to improve function than those with poor organization. Though we were not able to characterize it in the present study, we believe these observations suggest that there is a relationship between changes in function and changes in modular control which should be investigated in future studies.

Our evidence that the neuromuscular control strategies adopted before surgery are not permanent suggests that surgical intervention and rehabilitation have an influence on modular control strategy. These results contrast Shuman et al. [51] who found no change in the distribution of muscle weightings within modules after treatment in children with cerebral palsy. However, while the present study focuses on an orthopaedic condition rather than a neurological disorder, our results support the concepts presented by Ting et al. [30], who suggested that neuromuscular control patterns can be influenced by rehabilitation in individuals with spinal cord injury, stroke, and Parkinson's disease. Our results also support the findings of Roelker et al. [42] who found differences in motor modules between individuals with KOA and healthy, age-matched controls, indicating that the onset of KOA may alter neuromuscular control strategy. It remains unclear whether the changes in neural control observed in the present study resulted from changes in neuroanatomical structures (i.e. muscles, motor units, or even spinal/supraspinal structures) or are products of the neuromechanical interactions [30] caused by changes in the mechanical environment of the knee during surgery. Further, while some subjects presented increases in the number of modules or improvements in module organization over the time course of recovery, several participants decreased in number or worsened in organization (Table 4). These results may indicate that there may exist a neuromuscular control pattern, different from that of healthy older adults, which is more optimal for patients following TKA. Further investigation is needed to characterize "optimal control" in this population. Nonetheless, while we did not collect detailed data on the rehabilitation plans of the participants in the present pilot study, our results suggest that changes in neuromuscular control strategies, explored through motor modules, occur throughout the treatment and rehabilitation of an orthopaedic condition.

The plasticity of control in this population and the association between modular control and patient function at individual time points may provide a basis for modifying rehabilitation programs to improve functional outcomes after TKA. The association between better function and a higher number of modules before and 6-months after surgery and trends toward better average function in those with proper organization 24-months after surgery suggests that the relationships between function and module number or organization may be uncoupled. Our results before and 6-months after TKA support the findings of others, who have suggested that higher modular complexity indicates a higher number of independent motor strategies that can be used to complete a task [21,52], thus improving adaptability and function. Our work suggests that improvements in function in TKA patients may initially present as increases in the number of modules recruited during gait and that once individuals are capable of producing complex and discrete neuromuscular control patterns, further functional improvements may present as improvements in module organization. Further investigation is needed to determine the cause of the observed changes in modular control. The findings of this future work will help to determine if such motor control changes may be driven through rehabilitation programs in order to improve patient function.

This pilot study was based on a retrospective analysis of EMG data collected in a separate study, and as such, presented several limitations. Small sample sizes, particularly at the 24-month time point, may have prevented us from observing greater associations between modular control characteristics and function. The high BMI values, which are representative of the general TKA population, resulted in increased motion artifact during EMG collection and also limited the number of available trials for analysis. Further investigation is necessary to further characterize the relationship between changes in modular control characteristics and changes in function at the population level. A larger study may provide the statistical power needed to detect potential associations between modular complexity and functional outcomes. Additionally, EMGs were recorded from several muscles with similar functions (i.e., medial and lateral gastrocnemii, medial and lateral vasti) as a part of the original study design. This prevented us from analyzing modules involving other muscles that are important in gait (i.e., tibialis anterior, gluteus muscles). Given that the available EMG was collected from 3 groups of muscles with similar function (quadriceps, hamstrings, and plantarflexors), it seems appropriate that the maximum number of modules found was 3, with each typically dominated by the quadriceps, hamstrings, or plantarflexors. As a result, there was limited variation observed in the number of modules and our observations of modular control in this population were limited to these muscles. In future module studies, we plan to record a more diverse set of muscles.

## Conclusions

Some performance-based and patient-reported functional measures were related to modular complexity and organization. The number and organization of modules in individuals changed between the three testing time points in our study. We believe this new approach to evaluating motor control in individuals with TKA provides valuable insight into the plasticity of neural control strategies in patients and may help inform rehabilitation programs for patients demonstrating unsatisfactory outcomes. Future work will build on the findings of this pilot study and explore the influence of rehabilitation techniques, which target re-training neuromuscular control, on functional outcomes.

## Acknowledgments

The authors thank Dr. Jacqueline M. Lewis, Dr. Gregory M. Freisinger, Dr. Elena J. Caruthers, Rachel K. Hall, Elizabeth M. Leszcz, and Kathryn S. Blessinger for their contributions to data collection and processing. We also thank Drs. Jeffrey F. Granger, Andrew H. Glassman, and Matthew D. Beal, who performed the surgeries in this study, and Drs. Jason Payne and Alan Rogers, who performed the radiographic grading of OA severity.

## Author Contributions

**Conceptualization:** Rebekah R. Koehn, Sarah A. Roelker, Xueliang Pan, Laura C. Schmitt, Ajit M. W. Chaudhari, Robert A. Siston.

**Data curation:** Rebekah R. Koehn, Sarah A. Roelker, Xueliang Pan, Robert A. Siston.

**Formal analysis:** Rebekah R. Koehn, Sarah A. Roelker, Xueliang Pan, Robert A. Siston.

**Funding acquisition:** Xueliang Pan, Laura C. Schmitt, Ajit M. W. Chaudhari, Robert A. Siston.

**Investigation:** Rebekah R. Koehn, Sarah A. Roelker, Robert A. Siston.

**Methodology:** Rebekah R. Koehn, Sarah A. Roelker, Xueliang Pan, Laura C. Schmitt, Ajit M. W. Chaudhari, Robert A. Siston.

**Project administration:** Robert A. Siston.

**Resources:** Ajit M. W. Chaudhari, Robert A. Siston.

**Software:** Rebekah R. Koehn, Sarah A. Roelker, Xueliang Pan.

**Supervision:** Robert A. Siston.

**Visualization:** Rebekah R. Koehn, Sarah A. Roelker, Robert A. Siston.

**Writing – original draft:** Rebekah R. Koehn, Robert A. Siston.

**Writing – review & editing:** Rebekah R. Koehn, Sarah A. Roelker, Xueliang Pan, Laura C. Schmitt, Ajit M. W. Chaudhari, Robert A. Siston.

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
