## [Decision Letter · Decision Letter 0]

3 Feb 2022

PONE-D-21-39864IS MODULAR CONTROL RELATED TO FUNCTIONAL OUTCOMES IN INDIVIDUALS WITH KNEE OSTEOARTHRITIS AND FOLLOWING TOTAL KNEE ARTHROPLASTY?PLOS ONE

Dear Dr. Siston,

Thank you for submitting your manuscript to PLOS ONE. After careful consideration, we feel that it has merit but does not fully meet PLOS ONE’s publication criteria as it currently stands. Therefore, we invite you to submit a revised version of the manuscript that addresses the points raised during the review process.

We look forward to receiving your revised manuscript.

Kind regards,

Fatih Özden, PhD

Academic Editor

PLOS ONE

Journal Requirements:

Additional Editor Comments:

Dear Authors,

The reviewers has completed their comments. Please find their suggestion below. The manuscript should be revised regarding the comments. Reviewer 2 pointed out some important suggestions.

King Regards.

Reviewers' comments:

Reviewer's Responses to Questions

**Comments to the Author**

1. Is the manuscript technically sound, and do the data support the conclusions?

Reviewer #1: Yes

Reviewer #2: Partly

2. Has the statistical analysis been performed appropriately and rigorously? 

Reviewer #1: Yes

Reviewer #2: Yes

3. Have the authors made all data underlying the findings in their manuscript fully available?

Reviewer #1: Yes

Reviewer #2: Yes

4. Is the manuscript presented in an intelligible fashion and written in standard English?

Reviewer #1: Yes

Reviewer #2: Yes

5. Review Comments to the Author

Reviewer #1: Congratulations on the completion of your study and write up. Your study and findings on total knee arthroplasty and the factors that contribute to functional deficits post surgery are important and needed. Although your findings did not confirm your hypothesis, you still have good data that will contribute to the study, treatment, and practice of TKA. I recommend that this paper be accepted as it is free of errors, is written in standard English, has the data fully available, statistics are rigorous and appropriately performed, and the manuscript is technically sound.

There are two suggestions I have to improve your manuscript and also future studies. 1) Line 184 it was not clear why there was a dramatic decrease in subjects. I would add to the sentence "due to attrition or by pilot study design etc.". This would answer my questions much more easily instead of having to dig into the discussion or results. 2) In future studies, it would be better to use BIA or DEXA instead of BMI to characterize body composition or describe your participants. BMI is not a measure of body composition but is used because of it's ease and tradition in medicine and science. It is really a very poor measurement. Since you were already testing the subjects for functional measures, what would it hurt to also do a quick BIA analysis?

Nevertheless, well done on this study and the document.

Reviewer #2: General comments

This is a very well written and interesting paper, with proper statistical analyses. It investigated possible relationship between patient function and modular control during gait before and after total knee arthroplasty (TKA). Higher number of modules (muscle synergies) was associated to better performance-based and patient-reported function before and 6-months after surgery, although this was not verified 24-months after surgery. Moreover, results suggest that neuromuscular control strategies adopted before surgery are not permanent, which indicates that surgical intervention and rehabilitation have an influence on modular control strategy.

The topic may be relevant to a large number of researchers and clinical personnel working in motor control, orthopedics and rehabilitation fields. That are, however, some points that can be improved. Here are my comments on that.

The expression “36 participants (38 knees)” should be clarified. Moreover, authors should clarify how many participants repeated assessments and if some were not assessed before surgery but were assessed after. This is somehow plotted in Figures 3 and 4, but it would be easier for the reader to understand it if it was stated in the manuscript.

Lines 72-74: “Once individuals can produce discrete neuromuscular control patterns, further improvements may present as improved module organization.”. What do authors mean with “discrete neuromuscular control”?

In the third paragraph of the Introduction (lines 110-134), authors may also add the results reported in studies that investigated muscle synergies (motor modules) in related musculoskeletal conditions such as patellofemoral pain, which would help putting research in context.

Lines 164-166. Peripheral Nerve Stimulation electrodes are not ideal for electromyography. Moreover, the European Project SENIAM (currently the gold standard for EMG recommendations) showed that circular electrodes with a diameter of 10 mm are preferred. Dimensions of the electrodes used in the current study are far larger and this should be mentioned.

Tibialis anterior plays a crucial role during gait. EMG is usually recorded from this muscle during synergies analyses in gait and one of the synergies usually involves this muscle. Why did the authors decide to record other muscles with similar actions (e.g., lateral gastrocnemius and medial gastrocnemius) instead of using one of those EMG channels to record from tibialis anterior?

Lines 273-275: “The muscle weights of each module for each participant at each timepoint were compared to the averaged weights of healthy age-matched controls with the same number of modules using Pearson correlations.”. If healthy subjects can have 2 of 3 modules, how can you validate the use of number of modules as a correlate of patient function? This should at least be discussed. This is also related with my previous point of recording from Tibialis anterior.

Lines 358-367. What is the purpose of comparing tVAF from 1 to 8 synergies, if the number of extracted synergies was 2 or 3 synergies, depending on the subject?

Figure 3. It is hard to check all individual values, but it seems that some subjects go from proper to improper module organization. I think this was not discussed.

Section in lines 447-455. It could make sense to also describe results in terms of number of subjects that presented increased / decreased number of muscles, instead of just mentioning how many presented changes.

Minor comments

Lines 162-163: “Each participant performed a minimum of 5 over-ground walking trials.” There is no information on these trials: were they performed as overground walking or over a treadmill? Did the authors record heel strike moments or kinematics? How were beginning/end of strides detected?

Line 163: Missing comma in 1,500 Hz.

Line 165: Authors are encouraged to use the International System of Units (it should be cm or meters instead of inches).

Line 193: “Trials with excessive motion artifact were excluded”. Please clarify how this was done.

Line 196: “each trial contained 201 data points, accounting for every 0.5% of the gait cycle”. Please specify that each point corresponds to 0.5% of the gait cycle.

Line 247: For those readers who are not familiar with the concepts of Wmusc and Wsum, please briefly explain how these values are calculated.

What is the meaning of “clinically meaningful improvement”? Is this based on statistics (e.g., p-value < 0.01)?

Line 447. Based on the title of line 384, maybe this title should be “Hypothesis II: Changes in Modules”.

6. PLOS authors have the option to publish the peer review history of their article (what does this mean?). If published, this will include your full peer review and any attached files.

Reviewer #1: No

Reviewer #2: No

---

## [Author Response · Author response to Decision Letter 0]

15 Mar 2022

Editor’s Comments:

Journal Requirements:

We have made the following changes to the manuscript formatting:

1. All Level 1 headings are bold, 18pt font, and in sentence case.

2. All Level 2 headings are bold, 16pt font, and in sentence case.

3. All tables have been reformatted as editable, cell-based objects.

4. Funding and conflict of interest statements have been removed from the manuscript document.

5. Title page has been reformatted to comply with journal requirements.

Our Data Availability statement has updated and is included below:

“Data are available on Figshare.

Demographics data for all participants (.csv) and raw EMG, marker, and force plate data (.mat) for the participants before and after total knee arthroplasty are available at:

https://figshare.com/projects/Is_modular_control_related_to_functional_outcomes_in_individuals_with_knee_osteoarthritis_and_following_total_knee_arthroplasty_/132959

Raw EMG, marker, and force plate data (.mat) for the healthy participants are available at:

https://figshare.com/projects/Modular_Control_of_Walking_in_Unimpaired_Younger_and_Older_Adults_and_Individuals_with_Knee_Osteoarthritis/128774”

IRB approval has been clarified in the first sentence of the “Methods and materials” section:

“Prior to enrolling in the original longitudinal study (31), 36 individuals (38 knees) with medial compartment KOA (19/19 R/L) provided written informed consent. All study procedures were approved by The Ohio State University Institutional Review Board.”

Manuscript reference:

31. Freisinger GM, Hutter EE, Lewis J, Granger JF, Glassman AH, Beal MD, et al. Relationships between varus-valgus laxity of the severely osteoarthritic knee and gait, instability, clinical performance, and function. J Orthop Res. 2017;35(8):1644-52.

Additional Editor Comments:

Dear Authors,

The reviewers has completed their comments. Please find their suggestion below. The manuscript should be revised regarding the comments. Reviewer 2 pointed out some important suggestions.

King Regards.

 

Reviewers' comments:

Reviewer #1: Congratulations on the completion of your study and write up. Your study and findings on total knee arthroplasty and the factors that contribute to functional deficits post surgery are important and needed. Although your findings did not confirm your hypothesis, you still have good data that will contribute to the study, treatment, and practice of TKA. I recommend that this paper be accepted as it is free of errors, is written in standard English, has the data fully available, statistics are rigorous and appropriately performed, and the manuscript is technically sound.

There are two suggestions I have to improve your manuscript and also future studies.

[1.1]

1) Line 184 it was not clear why there was a dramatic decrease in subjects. I would add to the sentence "due to attrition or by pilot study design etc.". This would answer my questions much more easily instead of having to dig into the discussion or results.

The paragraph preceding this line (final paragraph of “Data collection”) has been clarified to provide more detailed explanation of data inclusion at each testing time point.

“Due to attrition and technical challenges in the original longitudinal study, all participants were not represented at all three data collection time points. Two participants received a TKA on both left and right knees and participated in the study twice. Hence 36 participants enrolled, yet 38 knees were included in the study. High BMI (e.g. 33.9 ± 5.1 kg/m2 before TKA), a common trait in individuals with osteoarthritis, caused soft tissue motion artifact in the EMG data in many of the trials. Due to this motion artifact, several trials were excluded from the study. Participants with fewer than 5 trials of usable EMG data at a particular testing time point were excluded from analysis at that time point. For these reasons and due to attrition, we were able to include data for 30 participants before surgery, 26 participants at 6-months post-TKA, and 13 participants at 24-months post-TKA (Table 1) in this pilot study. There were 8 participants with useable data at all three time points. All other participants had useable data at only two time points (11 before and 6-months after surgery, 2 before and 24-months after surgery, 2 at 6- and 24-months after surgery) or at only one time point (9 before surgery, 5 at 6-months after surgery, 1 at 24-months after surgery).”

[1.2]

2) In future studies, it would be better to use BIA or DEXA instead of BMI to characterize body composition or describe your participants. BMI is not a measure of body composition but is used because of it's ease and tradition in medicine and science. It is really a very poor measurement. Since you were already testing the subjects for functional measures, what would it hurt to also do a quick BIA analysis?

This pilot study was a retrospective analysis of previously collected data. BIA or DEXA were not part of the original study design. Thank you for the suggestion. We will consider BIA for future studies.

Nevertheless, well done on this study and the document.

Thank you for the kind words. We appreciate your review and comments.

Reviewer #2: General comments

This is a very well written and interesting paper, with proper statistical analyses. It investigated possible relationship between patient function and modular control during gait before and after total knee arthroplasty (TKA). Higher number of modules (muscle synergies) was associated to better performance-based and patient-reported function before and 6-months after surgery, although this was not verified 24-months after surgery. Moreover, results suggest that neuromuscular control strategies adopted before surgery are not permanent, which indicates that surgical intervention and rehabilitation have an influence on modular control strategy.

The topic may be relevant to a large number of researchers and clinical personnel working in motor control, orthopedics and rehabilitation fields. That are, however, some points that can be improved. Here are my comments on that.

[2.1]

The expression “36 participants (38 knees)” should be clarified. Moreover, authors should clarify how many participants repeated assessments and if some were not assessed before surgery but were assessed after. This is somehow plotted in Figures 3 and 4, but it would be easier for the reader to understand it if it was stated in the manuscript.

The number of participants vs. number of knees and details regarding repeated assessments of some participants have been clarified in the final paragraph of the “Data collection” subsection of “Methods and materials.” See response to comment 1.1, above.

[2.2]

Lines 72-74: “Once individuals can produce discrete neuromuscular control patterns, further improvements may present as improved module organization.”. What do authors mean with “discrete neuromuscular control”?

A higher number of motor modules represents more discrete patterns of control. We have removed this statement from the “Abstract” to avoid confusion:

“Subsequent improvements in function may present as improved module organization.”

[2.3]

In the third paragraph of the Introduction (lines 110-134), authors may also add the results reported in studies that investigated muscle synergies (motor modules) in related musculoskeletal conditions such as patellofemoral pain, which would help putting research in context.

We have added a statement highlighting muscle synergy research in populations with other knee-related musculoskeletal conditions:

“Recent literature has also indicated differences in motor modules between healthy adults and individuals with a variety of musculoskeletal knee conditions, including patellofemoral pain syndrome (26), anterior cruciate ligament deficiency (27), and KOA (28, 29).”

Manuscript references:

26. Lopes Ferreira C, Barroso FO, Torricelli D, Pons JL, Politti F, Lucareli PRG. Women with patellofemoral pain show altered motor coordination during lateral step down. J Biomech. 2020;110:109981.

27. Serrancolí G, Monllau JC, Font-Llagunes JM. Analysis of muscle synergies and activation-deactivation patterns in subjects with anterior cruciate ligament deficiency during walking. Clin Biomech (Bristol, Avon). 2016;31:65-73.

28. Kubota K, Hanawa H, Yokoyama M, Kita S, Hirata K, Fujino T, et al. Usefulness of Muscle Synergy Analysis in Individuals With Knee Osteoarthritis During Gait. IEEE Trans Neural Syst Rehabil Eng. 2021;29:239-48.

29. Roelker SA, Koehn RR, Caruthers EJ, Schmitt LC, Chaudhari AMW, Siston RA. Effects of age and knee osteoarthritis on the modular control of walking: A pilot study. PLoS One. 2021;16(12):e0261862.

[2.4]

Lines 164-166. Peripheral Nerve Stimulation electrodes are not ideal for electromyography. Moreover, the European Project SENIAM (currently the gold standard for EMG recommendations) showed that circular electrodes with a diameter of 10 mm are preferred. Dimensions of the electrodes used in the current study are far larger and this should be mentioned.

The sensors used in this study were comprised of 2 circular electrodes (with diameter 0.42 inches, which is approximately 10.6 mm) mounted to a rectangular adhesive foam with a width of 1.625 inches and a length of 3.25 inches. We have adjusted the language to more clearly report the relevant electrode parameters (in SI units, in accordance with comment 2.12).

“Surface EMG data were collected at 1,500 Hz (Telemyo DTS System, Noraxon, Scottsdale, AZ) from 16 pre-gelled Ag/AgCl dual-electrodes (Model A10011, 10.592 mm sensor diameter, 40 mm inter-electrode distance; Vermed, Buffalo, NY) affixed over the bellies of 8 lower extremity muscles, bilaterally…”

While these sensors are marketed for potential use in peripheral nerve stimulation, they have been used to collect EMG data in many previously published studies (Caruthers et al., 2018; Caruthers et al., 2020; Caruthers et al., 2016; Chaudhari et al., 2014; Chaudhari et al., 2020; Chaudhari et al., 2019; Freisinger et al., 2017; Jamison et al., 2013; Jamison et al., 2012; Lewis et al., 2015; Roelker et al., 2021; Thoma et al., 2016). These sensors comply with SENIAM recommendations for shape (round), size (10 mm diameter), material (pre-gelled Ag/AgCl), and sensor construction (fixed inter-electrode distance, lightweight material). While the inter-electrode distance of 40 mm is larger than the SENIAM-recommended 20 mm, inter-electrode distances of 20 mm and 40 mm have been found to result in similar EMG profiles (Beck et al., 2009). Thus, we believe these sensors were suitable for our study and do not present a limitation to our findings.

Comment references:

Beck, T. W., Housh, T. J., Cramer, J. T., & Weir, J. P. (2009). The effects of interelectrode distance over the innervation zone and normalization on the electromyographic amplitude and mean power frequency versus concentric, eccentric, and isometric torque relationships for the vastus lateralis muscle. J Electromyogr Kinesiol, 19(2), 219-231. https://doi.org/10.1016/j.jelekin.2007.07.007

Caruthers, E. J., Oxendale, K. K., Lewis, J. M., Chaudhari, A. M. W., Schmitt, L. C., Best, T. M., & Siston, R. A. (2018). Forces Generated by Vastus Lateralis and Vastus Medialis Decrease with Increasing Stair Descent Speed. Ann Biomed Eng, 46(4), 579-589. https://doi.org/10.1007/s10439-018-1979-9

Caruthers, E. J., Schneider, G., Schmitt, L. C., Chaudhari, A. M. W., & Siston, R. A. (2020). What are the effects of simulated muscle weakness on the sit-to-stand transfer? Comput Methods Biomech Biomed Engin, 23(11), 765-772. https://doi.org/10.1080/10255842.2020.1764544

Caruthers, E. J., Thompson, J. A., Chaudhari, A. M., Schmitt, L. C., Best, T. M., Saul, K. R., & Siston, R. A. (2016). Muscle Forces and Their Contributions to Vertical and Horizontal Acceleration of the Center of Mass During Sit-to-Stand Transfer in Young, Healthy Adults. J Appl Biomech, 32(5), 487-503. https://doi.org/10.1123/jab.2015-0291

Chaudhari, A. M., Jamison, S. T., McNally, M. P., Pan, X., & Schmitt, L. C. (2014). Hip adductor activations during run-to-cut manoeuvres in compression shorts: implications for return to sport after groin injury. J Sports Sci, 32(14), 1333-1340. https://doi.org/10.1080/02640414.2014.889849

Chaudhari, A. M. W., MR, V. A. N. H., Monfort, S. M., Pan, X., Oñate, J. A., & Best, T. M. (2020). Reducing Core Stability Influences Lower Extremity Biomechanics in Novice Runners. Med Sci Sports Exerc, 52(6), 1347-1353. https://doi.org/10.1249/mss.0000000000002254

Chaudhari, A. M. W., Schmitt, L. C., Freisinger, G. M., Lewis, J. M., Hutter, E. E., Pan, X., & Siston, R. A. (2019). Perceived Instability Is Associated With Strength and Pain, Not Frontal Knee Laxity, in Patients With Advanced Knee Osteoarthritis. J Orthop Sports Phys Ther, 49(7), 513-517. https://doi.org/10.2519/jospt.2019.8619

Freisinger, G. M., Hutter, E. E., Lewis, J., Granger, J. F., Glassman, A. H., Beal, M. D., . . . Chaudhari, A. M. W. (2017). Relationships between varus-valgus laxity of the severely osteoarthritic knee and gait, instability, clinical performance, and function. J Orthop Res, 35(8), 1644-1652. https://doi.org/10.1002/jor.23447

Jamison, S. T., McNally, M. P., Schmitt, L. C., & Chaudhari, A. M. (2013). The effects of core muscle activation on dynamic trunk position and knee abduction moments: implications for ACL injury. J Biomech, 46(13), 2236-2241. https://doi.org/10.1016/j.jbiomech.2013.06.021

Jamison, S. T., Pan, X., & Chaudhari, A. M. (2012). Knee moments during run-to-cut maneuvers are associated with lateral trunk positioning. J Biomech, 45(11), 1881-1885. https://doi.org/10.1016/j.jbiomech.2012.05.031

Lewis, J., Freisinger, G., Pan, X., Siston, R., Schmitt, L., & Chaudhari, A. (2015). Changes in lower extremity peak angles, moments and muscle activations during stair climbing at different speeds. J Electromyogr Kinesiol, 25(6), 982-989. https://doi.org/10.1016/j.jelekin.2015.07.011

Roelker, S. A., Koehn, R. R., Caruthers, E. J., Schmitt, L. C., Chaudhari, A. M. W., & Siston, R. A. (2021). Effects of age and knee osteoarthritis on the modular control of walking: A pilot study. PLoS One, 16(12), e0261862. https://doi.org/10.1371/journal.pone.0261862

Thoma, L. M., McNally, M. P., Chaudhari, A. M., Flanigan, D. C., Best, T. M., Siston, R. A., & Schmitt, L. C. (2016). Muscle co-contraction during gait in individuals with articular cartilage defects in the knee. Gait Posture, 48, 68-73. https://doi.org/10.1016/j.gaitpost.2016.04.021

[2.5]

Tibialis anterior plays a crucial role during gait. EMG is usually recorded from this muscle during synergies analyses in gait and one of the synergies usually involves this muscle. Why did the authors decide to record other muscles with similar actions (e.g., lateral gastrocnemius and medial gastrocnemius) instead of using one of those EMG channels to record from tibialis anterior?

This pilot study was a retrospective analysis of previously collected data. Module analysis was not foreseen when the original study (Freisinger et al., 2017) was designed. The medial and lateral gastrocnemius were recorded with the original intention of examining co-contraction indices of the muscles surrounding the knee, thus EMG collection was limited to muscles which actuate the knee. This pilot study and will be used to inform subsequent module studies, in which we plan to record from the tibialis anterior. We added text to address this as a limitation at the end of the “Discussion” section:

“Additionally, EMGs were recorded from several muscles with similar functions (i.e., medial and lateral gastrocnemii, medial and lateral vasti) as a part of the original study design. This prevented us from analyzing modules involving other muscles that are important in gait (i.e., tibialis anterior, gluteus muscles). Given that the available EMG was collected from 3 groups of muscles with similar function (quadriceps, hamstrings, and plantarflexors), it seems appropriate that the maximum number of modules found was 3, with each typically dominated by the quadriceps, hamstrings, or plantarflexors. As a result, there was limited variation observed in the number of modules and our observations of modular control in this population were limited to these muscles. In future module studies, we plan to record a more diverse set of muscles.”

Comment reference:

Freisinger, G. M., Hutter, E. E., Lewis, J., Granger, J. F., Glassman, A. H., Beal, M. D., . . . Chaudhari, A. M. W. (2017). Relationships between varus-valgus laxity of the severely osteoarthritic knee and gait, instability, clinical performance, and function. J Orthop Res, 35(8), 1644-1652. https://doi.org/10.1002/jor.23447

[2.6]

Lines 273-275: “The muscle weights of each module for each participant at each timepoint were compared to the averaged weights of healthy age-matched controls with the same number of modules using Pearson correlations.”. If healthy subjects can have 2 of 3 modules, how can you validate the use of number of modules as a correlate of patient function? This should at least be discussed. This is also related with my previous point of recording from Tibialis anterior.

We have added a paragraph to the “Discussion” section to justify the use of module number as a correlate of patient function and of our healthy control group selection:

“This study was motivated by previous work which has found that a higher number of modules is related to better function (21, 24, 25) and by our own previous work (29) where we observed decreases in the number of modules in healthy older adults and older adults with knee osteoarthritis (KOA) compared to healthy younger adults. In all of these studies, healthy older adults have been found to demonstrate a range in the number of motor modules, from as low as 2 to as high as 6, depending on the extraction method and EMG sensor pattern (21, 24, 25, 29) . In the current study, we used the healthy subjects from Roelker et al. (29) for direct comparison of the number of modules between individuals with TKA and healthy adults using the same module extraction methods and experimental configuration. In Roelker et al. (29), we observed that healthy older adults demonstrated only 2 or 3 modules despite demonstrating similar self-selected walking speed to healthy younger adults, who demonstrated up to 4 modules. Given that there were no differences in self-selected walking speed between the healthy younger and healthy older adults, we believe this healthy older adult cohort was a suitable control group for the current study.”

Further, including additional muscles with more unique functions in gait, such as the tibialis anterior, would likely result in a higher number of synergies and may provide greater variation in the “number of modules” metric for analysis. We have added a statement regarding this point at the end of the “Discussion” section (see response to comment 2.5).

Manuscript references:

21. Ardestani MM, Malloy P, Nam D, Rosenberg AG, Wimmer MA. TKA patients with unsatisfying knee function show changes in neuromotor synergy pattern but not joint biomechanics. J Electromyogr Kinesiol. 2017;37:90-100.

24. Clark DJ, Ting LH, Zajac FE, Neptune RR, Kautz SA. Merging of healthy motor modules predicts reduced locomotor performance and muscle coordination complexity post-stroke. J Neurophysiol. 2010;103(2):844-57.

25. Hayes HB, Chvatal SA, French MA, Ting LH, Trumbower RD. Neuromuscular constraints on muscle coordination during overground walking in persons with chronic incomplete spinal cord injury. Clin Neurophysiol. 2014;125(10):2024-35.

29. Roelker SA, Koehn RR, Caruthers EJ, Schmitt LC, Chaudhari AMW, Siston RA. Effects of age and knee osteoarthritis on the modular control of walking: A pilot study. PLoS One. 2021;16(12):e0261862.

[2.7]

Lines 358-367. What is the purpose of comparing tVAF from 1 to 8 synergies, if the number of extracted synergies was 2 or 3 synergies, depending on the subject?

While the number of motor modules is often used as a measure of neuromuscular control complexity, complexity has also been quantified using tVAF values in previous literature (Jacobs et al., 2018; Shuman et al., 2018; Steele et al., 2015). Given that all participants demonstrated only 2 or 3 modules in the current study, tVAF provided greater resolution to investigate changes in control complexity over time. We now comment on this in the “Discussion” section:

“While the number of motor modules is often used as a measure of neuromuscular control complexity, complexity has also been quantified using tVAF values in previous literature (44-46). Given that all participants demonstrated only 2 or 3 modules in the current study, tVAF provided greater resolution to investigate changes in control complexity over time.”

Manuscript and comment references:

44. Steele KM, Rozumalski A, Schwartz MH. Muscle synergies and complexity of neuromuscular control during gait in cerebral palsy. Dev Med Child Neurol. 2015;57(12):1176-82.

45. Jacobs DA, Koller JR, Steele KM, Ferris DP. Motor modules during adaptation to walking in a powered ankle exoskeleton. J Neuroeng Rehabil. 2018;15(1):2.

46. Shuman BR, Goudriaan M, Desloovere K, Schwartz MH, Steele KM. Associations Between Muscle Synergies and Treatment Outcomes in Cerebral Palsy Are Robust Across Clinical Centers. Arch Phys Med Rehabil. 2018;99(11):2175-82.

[2.8]

Figure 3. It is hard to check all individual values, but it seems that some subjects go from proper to improper module organization. I think this was not discussed.

A tally of the number of subjects who improved/worsened in organization or increased/decreased in number of modules are presented in Table 4. We have added a statement to address those subjects who worsened in organization or decreased in number of modules in the “Discussion”:

“Further, while some subjects presented increases in the number of modules or improvements in module organization over the time course of recovery, several participants decreased in number or worsened in organization (Table 4). These results may indicate that there may exist a neuromuscular control pattern, different from that of healthy older adults, which is more optimal for patients following TKA. Further investigation is needed to characterize “optimal control” in this population.”

[2.9]

Section in lines 447-455. It could make sense to also describe results in terms of number of subjects that presented increased / decreased number of muscles, instead of just mentioning how many presented changes.

This point is addressed in the response to comment 2.8, above.

Minor comments

[2.10]

Lines 162-163: “Each participant performed a minimum of 5 over-ground walking trials.” There is no information on these trials: were they performed as overground walking or over a treadmill? Did the authors record heel strike moments or kinematics? How were beginning/end of strides detected?

Gait trials were performed as over-ground walking trials. Kinematic and kinetic data were not analyzed in this study. Details regarding the separation of gait cycles have been clarified:

“Force plate data were collected at 1,500 Hz and were used to identify heel-strike-to-heel-strike gait cycle timing. Motion capture data were also collected but were not used in this study. Details of motion capture collection and analysis can be found in Freisinger et al. (31).”

Manuscript reference:

31. Freisinger GM, Hutter EE, Lewis J, Granger JF, Glassman AH, Beal MD, et al. Relationships between varus-valgus laxity of the severely osteoarthritic knee and gait, instability, clinical performance, and function. J Orthop Res. 2017;35(8):1644-52.

[2.11]

Line 163: Missing comma in 1,500 Hz.

A comma has been added (second paragraph of “Data collection”).

[2.12]

Line 165: Authors are encouraged to use the International System of Units (it should be cm or meters instead of inches).

We have adjusted the electrode measurements to SI units. See response to comment 2.4.

[2.13]

Line 193: “Trials with excessive motion artifact were excluded”. Please clarify how this was done.

We have clarified that motion artifact was identified visually:

“All trials were examined visually, and those with missing channels, gaps in EMG data, or excessive motion artifact were excluded.”

[2.14]

Line 196: “each trial contained 201 data points, accounting for every 0.5% of the gait cycle”. Please specify that each point corresponds to 0.5% of the gait cycle.

“Linear envelopes were formed by discretizing the data such that each trial contained 201 data points, such that each point corresponds to 0.5% of the gait cycle, concatenating all available gait cycles, and normalizing each muscle first to the maximum value across all gait cycles and then to unit variance (40).”

Manuscript reference:

40. Torres-Oviedo G, Ting LH. Muscle synergies characterizing human postural responses. J Neurophysiol. 2007;98(4):2144-56.

[2.15]

Line 247: For those readers who are not familiar with the concepts of Wmusc and Wsum, please briefly explain how these values are calculated.

The definitions of Wmusc and Wsum and the details for finding these values are provided under “Materials and methods.” We have adjusted the text for clarity:

“To characterize modular organization, we adapted definitions of organization characteristics, Wmusc and Wsum, from Hayes et al. (25). Wmusc is defined as the number of significantly active muscles in each module, and Wsum is defined as the sum of the weights of the significantly active muscles in each module. In Hayes et al. (25) a muscle was considered significantly active if the 95% CI of weights for the 500 bootstrapped gait cycles did not contain 0. However, participants in the current study often had confidence intervals with a lower bound greater than zero but a very small upper bound. Under the original definition from Hayes et al. (reference), all modules for all participants in the TKA cohort had a Wmusc of 7 or 8, meaning all muscles were significantly active in all modules. To uncover more subtle differences in Wmusc and Wsum between participants in our cohort, we adapted the definition of “significantly active,” such that a muscle was considered significantly active if the 95% CI of weights did not contain 0 and the upper bound was greater than 0.25. Modules from each of the 500 samples were sorted using a k-means algorithm to ensure like-modules were grouped prior to determining the 95% CIs for the weights (reference). The significantly active muscles in each module were counted (Wmusc) and the weights of these muscles were added (Wsum).”

Manuscript reference:

25. Hayes HB, Chvatal SA, French MA, Ting LH, Trumbower RD. Neuromuscular constraints on muscle coordination during overground walking in persons with chronic incomplete spinal cord injury. Clin Neurophysiol. 2014;125(10):2024-35.

[2.16]

What is the meaning of “clinically meaningful improvement”? Is this based on statistics (e.g., p-value < 0.01)?

Clinically meaningful improvement is represented by change in function greater than or equal to the minimum detectable changes defined in previous literature and described in the “Materials and methods” section. We have added text to clarify this:

“Clinically meaningful changes in functional performance

Clinically meaningful differences in performance-based and self-reported measures were evaluated based on minimum detectable changes (MDCs), as defined by previously reported 90% CIs from representative populations. Improvements were indicated by decreases in the time to complete SCT (� 1.9 seconds; (43)) and TUG (� 2.49 seconds; (36)), increases in the distance walked during 6MW (� 61.34 meters; (36)), and increases in all KOOS subscales (� 10 points; (37)).”

Manuscript references:

36. Kennedy DM, Stratford PW, Wessel J, Gollish JD, Penney D. Assessing stability and change of four performance measures: a longitudinal study evaluating outcome following total hip and knee arthroplasty. BMC Musculoskelet Disord. 2005;6:3.

37. Roos EM, Lohmander LS. The Knee injury and Osteoarthritis Outcome Score (KOOS): from joint injury to osteoarthritis. Health Qual Life Outcomes. 2003;1:64.

43. Almeida GJ, Schroeder CA, Gil AB, Fitzgerald GK, Piva SR. Interrater reliability and validity of the stair ascend/descend test in subjects with total knee arthroplasty. Arch Phys Med Rehabil. 2010;91(6):932-8.

[2.17]

Line 447. Based on the title of line 384, maybe this title should be “Hypothesis II: Changes in Modules”.

This section of text (titled “Secondary analysis: Changes in modules”) describes the results of our secondary analysis. The results regarding Hypothesis II are described in the previous section (titled “Hypothesis II: Function and module organization”).

 

Response to Reviewers References:

Ardestani, M. M., Malloy, P., Nam, D., Rosenberg, A. G., & Wimmer, M. A. (2017). TKA patients with unsatisfying knee function show changes in neuromotor synergy pattern but not joint biomechanics. J Electromyogr Kinesiol, 37, 90-100. https://doi.org/10.1016/j.jelekin.2017.09.006

Beck, T. W., Housh, T. J., Cramer, J. T., & Weir, J. P. (2009). The effects of interelectrode distance over the innervation zone and normalization on the electromyographic amplitude and mean power frequency versus concentric, eccentric, and isometric torque relationships for the vastus lateralis muscle. J Electromyogr Kinesiol, 19(2), 219-231. https://doi.org/10.1016/j.jelekin.2007.07.007

Caruthers, E. J., Oxendale, K. K., Lewis, J. M., Chaudhari, A. M. W., Schmitt, L. C., Best, T. M., & Siston, R. A. (2018). Forces Generated by Vastus Lateralis and Vastus Medialis Decrease with Increasing Stair Descent Speed. Ann Biomed Eng, 46(4), 579-589. https://doi.org/10.1007/s10439-018-1979-9

Caruthers, E. J., Schneider, G., Schmitt, L. C., Chaudhari, A. M. W., & Siston, R. A. (2020). What are the effects of simulated muscle weakness on the sit-to-stand transfer? Comput Methods Biomech Biomed Engin, 23(11), 765-772. https://doi.org/10.1080/10255842.2020.1764544

Caruthers, E. J., Thompson, J. A., Chaudhari, A. M., Schmitt, L. C., Best, T. M., Saul, K. R., & Siston, R. A. (2016). Muscle Forces and Their Contributions to Vertical and Horizontal Acceleration of the Center of Mass During Sit-to-Stand Transfer in Young, Healthy Adults. J Appl Biomech, 32(5), 487-503. https://doi.org/10.1123/jab.2015-0291

Chaudhari, A. M., Jamison, S. T., McNally, M. P., Pan, X., & Schmitt, L. C. (2014). Hip adductor activations during run-to-cut manoeuvres in compression shorts: implications for return to sport after groin injury. J Sports Sci, 32(14), 1333-1340. https://doi.org/10.1080/02640414.2014.889849

Chaudhari, A. M. W., MR, V. A. N. H., Monfort, S. M., Pan, X., Oñate, J. A., & Best, T. M. (2020). Reducing Core Stability Influences Lower Extremity Biomechanics in Novice Runners. Med Sci Sports Exerc, 52(6), 1347-1353. https://doi.org/10.1249/mss.0000000000002254

Chaudhari, A. M. W., Schmitt, L. C., Freisinger, G. M., Lewis, J. M., Hutter, E. E., Pan, X., & Siston, R. A. (2019). Perceived Instability Is Associated With Strength and Pain, Not Frontal Knee Laxity, in Patients With Advanced Knee Osteoarthritis. J Orthop Sports Phys Ther, 49(7), 513-517. https://doi.org/10.2519/jospt.2019.8619

Clark, D. J., Ting, L. H., Zajac, F. E., Neptune, R. R., & Kautz, S. A. (2010). Merging of healthy motor modules predicts reduced locomotor performance and muscle coordination complexity post-stroke. J Neurophysiol, 103(2), 844-857. https://doi.org/10.1152/jn.00825.2009

Freisinger, G. M., Hutter, E. E., Lewis, J., Granger, J. F., Glassman, A. H., Beal, M. D., . . . Chaudhari, A. M. W. (2017). Relationships between varus-valgus laxity of the severely osteoarthritic knee and gait, instability, clinical performance, and function. J Orthop Res, 35(8), 1644-1652. https://doi.org/10.1002/jor.23447

Hayes, H. B., Chvatal, S. A., French, M. A., Ting, L. H., & Trumbower, R. D. (2014). Neuromuscular constraints on muscle coordination during overground walking in persons with chronic incomplete spinal cord injury. Clin Neurophysiol, 125(10), 2024-2035. https://doi.org/10.1016/j.clinph.2014.02.001

Jacobs, D. A., Koller, J. R., Steele, K. M., & Ferris, D. P. (2018). Motor modules during adaptation to walking in a powered ankle exoskeleton. J Neuroeng Rehabil, 15(1), 2. https://doi.org/10.1186/s12984-017-0343-x

Jamison, S. T., McNally, M. P., Schmitt, L. C., & Chaudhari, A. M. (2013). The effects of core muscle activation on dynamic trunk position and knee abduction moments: implications for ACL injury. J Biomech, 46(13), 2236-2241. https://doi.org/10.1016/j.jbiomech.2013.06.021

Jamison, S. T., Pan, X., & Chaudhari, A. M. (2012). Knee moments during run-to-cut maneuvers are associated with lateral trunk positioning. J Biomech, 45(11), 1881-1885. https://doi.org/10.1016/j.jbiomech.2012.05.031

Lewis, J., Freisinger, G., Pan, X., Siston, R., Schmitt, L., & Chaudhari, A. (2015). Changes in lower extremity peak angles, moments and muscle activations during stair climbing at different speeds. J Electromyogr Kinesiol, 25(6), 982-989. https://doi.org/10.1016/j.jelekin.2015.07.011

Roelker, S. A., Koehn, R. R., Caruthers, E. J., Schmitt, L. C., Chaudhari, A. M. W., & Siston, R. A. (2021). Effects of age and knee osteoarthritis on the modular control of walking: A pilot study. PLoS One, 16(12), e0261862. https://doi.org/10.1371/journal.pone.0261862

Shuman, B. R., Goudriaan, M., Desloovere, K., Schwartz, M. H., & Steele, K. M. (2018). Associations Between Muscle Synergies and Treatment Outcomes in Cerebral Palsy Are Robust Across Clinical Centers. Arch Phys Med Rehabil, 99(11), 2175-2182. https://doi.org/10.1016/j.apmr.2018.03.006

Steele, K. M., Rozumalski, A., & Schwartz, M. H. (2015). Muscle synergies and complexity of neuromuscular control during gait in cerebral palsy. Dev Med Child Neurol, 57(12), 1176-1182. https://doi.org/10.1111/dmcn.12826

Thoma, L. M., McNally, M. P., Chaudhari, A. M., Flanigan, D. C., Best, T. M., Siston, R. A., & Schmitt, L. C. (2016). Muscle co-contraction during gait in individuals with articular cartilage defects in the knee. Gait Posture, 48, 68-73. https://doi.org/10.1016/j.gaitpost.2016.04.021

---

## [Decision Letter · Decision Letter 1]

7 Apr 2022

IS MODULAR CONTROL RELATED TO FUNCTIONAL OUTCOMES IN INDIVIDUALS WITH KNEE OSTEOARTHRITIS AND FOLLOWING TOTAL KNEE ARTHROPLASTY?

PONE-D-21-39864R1

Dear Dr. Siston,

We’re pleased to inform you that your manuscript has been judged scientifically suitable for publication and will be formally accepted for publication once it meets all outstanding technical requirements.

Kind regards,

Fatih Özden, PhD

Academic Editor

PLOS ONE

Additional Editor Comments (optional):

-

Reviewers' comments:

Reviewer's Responses to Questions

**Comments to the Author**

1. If the authors have adequately addressed your comments raised in a previous round of review and you feel that this manuscript is now acceptable for publication, you may indicate that here to bypass the “Comments to the Author” section, enter your conflict of interest statement in the “Confidential to Editor” section, and submit your "Accept" recommendation.

Reviewer #1: All comments have been addressed

Reviewer #2: All comments have been addressed

2. Is the manuscript technically sound, and do the data support the conclusions?

Reviewer #1: Yes

Reviewer #2: Yes

3. Has the statistical analysis been performed appropriately and rigorously? 

Reviewer #1: Yes

Reviewer #2: Yes

4. Have the authors made all data underlying the findings in their manuscript fully available?

Reviewer #1: Yes

Reviewer #2: Yes

5. Is the manuscript presented in an intelligible fashion and written in standard English?

Reviewer #1: Yes

Reviewer #2: (No Response)

6. Review Comments to the Author

Reviewer #1: Thank you for your clarification about my initial comments. I am satisfied with your efforts. Good luck with the publication.

Reviewer #2: I want to thank the authors for addressing my previous comments. In my opinion, the current version of the manuscript makes a valuable contribution to the field.

7. PLOS authors have the option to publish the peer review history of their article (what does this mean?). If published, this will include your full peer review and any attached files.

Reviewer #1: **Yes: **Kathryn Rosie Lanphere

Reviewer #2: No

---

## [Editor Report · Acceptance letter]

14 Apr 2022

PONE-D-21-39864R1 

Is modular control related to functional outcomes in individuals with knee osteoarthritis and following total knee arthroplasty? 

Dear Dr. Siston:

I'm pleased to inform you that your manuscript has been deemed suitable for publication in PLOS ONE. Congratulations! Your manuscript is now with our production department. 

Kind regards, 

on behalf of

Dr. Fatih Özden 

Academic Editor

PLOS ONE